# Cellular macromolecules-tethered DNA walking indexing to explore nanoenvironments of chromatin modifications

Feng Chen[1,4], Min Bai[1,4], Xiaowen Cao[1], Jing Xue[1], Yue Zhao[1], Na Wu[1], Lei Wang[2], Dexin Zhang[3] & Yongxi Zhao ![ORCID][1✉]

Exploring spatial organization and relationship of diverse biomolecules within cellular nanoenvironments is important to elucidate the fundamental processes of life. However, it remains methodologically challenging. Herein, we report a molecular recognition mechanism cellular macromolecules-tethered DNA walking indexing (Cell-TALKING) to probe the nanoenvironments containing diverse chromatin modifications. As an example, we characterize the nanoenvironments of three DNA modifications around one histone post-translational modification (PTM). These DNA modifications in fixed cells are labeled with respective DNA barcoding probes, and then the PTM site is tethered with a DNA walking probe. Cell-TALKING can continuously produce cleavage records of any barcoding probes nearby the walking probe. New 3'-OH ends are generated on the cleaved barcoding probes to induce DNA amplification for downstream detections. Combining fluorescence imaging, we identify various combinatorial chromatin modifications and investigate their dynamic changes during cell cycles. We also explore the nanoenvironments in different cancer cell lines and clinical specimens. In principle, using high-throughput sequencing instead of fluorescence imaging may allow the detection of complex cellular nanoenvironments containing tens of biomolecules such as transcription factors.

[1] Institute of Analytical Chemistry and Instrument for Life Science, The Key Laboratory of Biomedical Information Engineering of Ministry of Education, School of Life Science and Technology, Xi'an Jiaotong University, Xi'an, Shaanxi, PR China. [2] Department of Thoracic Surgery, Tangdu Hospital, Air Force Medical University, Xi'an, Shaanxi, PR China. [3] Department of Respiratory Medicine, The Second Affiliated Hospital of Medical College, Xi'an Jiaotong University, Xi'an, Shaanxi, PR China. [4] These authors contributed equally: Feng Chen, Min Bai. ✉email: yxzhao@mail.xjtu.edu.cn

Spatial organization and relationship of diverse biomolecules within cellular environments controls essential biological functions and underpins the fundamental processes of life. For instance, chromatin is made up of basic structural subunits called nucleosomes (~10 nm) and is extensively decorated by diverse DNA modifications and histone posttranslational modifications (PTMs)[1]. These chemical modifications can be recognized by specialized proteins or protein complexes. They have been shown to act as critical regulators of gene expression, genome replication, and/or chromatin architecture in organismal development and human health. 5-Methylcytosine (5mC) is the most common DNA modification in mammalian genomes and is associated with various diseases such as cancer[2]. Successive oxidation of 5mC to 5-hydroxymethylcytosine (5hmC) and 5-formylcytosine (5fC) by ten-eleven translocation enzymes promotes the active DNA demethylation. Accumulated evidences have shown that these oxidized cytosine modifications also play important roles in gene regulation[3–7]. In addition to cytosine, thymine can be oxidized to 5-hydroxymethyluracil (5hmU) and 5-formyluracil (5fU)[8,9]. These thymine modifications are commonly known as DNA oxidative lesions, and 5hmU may alter the binding of some transcription factors (TFs) and chromatin remodeling proteins in mouse embryonic stem cells[8]. On the other hand, histones can be modified by diverse PTMs, ~15 known chemical moieties and hundreds of modifiable amino acid residues. The best-characterized histone PTMs include methylation, acetylation, ubiquitination, and phosphorylation. Specific histone PTMs are associated with active or silenced genomic regions, and hence play important roles in chromatin states and gene regulation[10,11]. For instance, lysine methylation modifications such as H3K4me1 and H3K27me3 are correlated with either gene activation or gene silencing, which depends on the types of methylation and/or the modification status of proximal residues. Furthermore, the co-occurrence of two chromatin modifications at the same genomic region or between two close regions has been confirmed[10,12,13], implying their spatial relationship and potential crosstalk[1,14]. The diversity of chromatin modifications underscores the demand to probe their complex compositions and spatial relationships at nanoscale (e.g., <20 nm) termed nanoenvironments. For example, we here define many DNA modifications around individual histone PTM as one type of nanoenvironments of chromatin modifications. It may reveal the potential interactions between their reader proteins in a spatio-temporal manner, and help elucidate complex epigenetic function in cellular events and pathological processes. However, this issue remains scarcely explored.

In the past decade, the chromatin immunoprecipitation (ChIP) deep sequencing and its derivatives have been well established to analyze genomic positioning of histone PTMs[10,12,15]. Different sequencing strategies have also been developed for genome-wide profiling of DNA modifications including 5mC, 5hmC, 5fC, or 5hmU[4–7,9,16]. These methods reported chemical or chemoenzymatic labeling of modified bases, and revealed their important biological functions. ChIP can be combined with bisulfite sequencing to interrogate the genomic distribution relationship between 5mC and a histone PTM[17,18]. This method is limited to 5mC only and is unable to assess other DNA modifications. Sequencing-based methods have deepened our understanding of the functions and roles of chromatin modifications. However, these methods rely on substantial starting materials and produce cell population-averaged information. They are unable to probe subnuclear distributions and spatial organization of diverse chromatin modifications.

Microscopy imaging can directly detect spatial positioning and distribution of biomolecules inside individual cells. We and others have demonstrated fluorescence visualization of single-cell DNA modifications or histone PTMs, revealing their cell-type-dependent distribution signatures with remarkable cell-to-cell heterogeneity[19–23]. We have reported specific thiophosphorylation of 5hmU against its structurally similar analogs such as 5hmC and 5fU, and achieved differentiated imaging of 5hmU and 5hmC in the same cells[20]. Aforementioned imaging methods could be routinely expanded for parallel detection of three or more spatially independent modifications by designing orthogonal probe sets. Yet they are unable to probe spatial proximity relationships between two or more different modifications within nanoenvironments. In addition, DNA proximity ligation assay (PLA) has been developed to record the colocalization of two adjacent proteins, genomic regions, or chemical modifications by ligating sequences from two DNA probes[23–26]. However, the probes are consumed in the detection of an individual one-to-one combination or relationship. This makes PLA difficult to probe complex relationships of diverse adjacent target molecules. DNA nanotechnology has been rapidly progressing during the past decade. It contributed major breakthroughs toward programmable and nanoscale molecular assembly, positioning, and organization[27–30]. Recently, we and others have developed various DNA nanostructured probes or machines for proximity recognition, assembly, or disassembly[31–36]. In particular, some of these DNA machines can implement sequential assembly–disassembly cycles driven by DNA digestion or strand displacement[31,33–35]. During this reaction process, an individual wallking molecule can continuously scan any nearby track molecules within a local environment. These advancements may meet the challenge of probing the nanoenvironmenst containing diverse biomolecules inside single cells.

Here, we reported a cellular macromolecules-tethered DNA walking indexing (Cell-TALKING) technique capable of probing cellular nanoenvironments containing diverse biomolecules. This mechanism can continuously produce DNA proximity cleavage records of any interested biomolecules (1, 2, 3,…N) nearby individual target sites (T) in fixed cells (Fig. 1a). Respective DNA 3′-OH end is generated to activate the barcode through rolling circle amplification (RCA) and polymerase chain reaction (PCR). Downstream detection techniques such as fluorescence imaging and DNA sequencing can be used to detect these corresponding amplicons. As an example, we used Cell-TALKING to porbe the nanoenvironment of three DNA modifications (5hmU, 5hmC, and 5fU) around individual histone PTMs combined with confocal fluorescence imaging (Fig. 1b). In brief, these DNA modifications were labeled with respective DNA barcoding probes (BPs), and the PTM site was tethered with a DNA walking probe (WP). After full reactions of Cell-TALKING, the detected modification sites can be visualized by RCA-improved fluorescence imaging. Diverse combinatorial patterns or relationships of these chromatin modifications were identified, and the information of subnuclear distributions were provided with single-site resolution. We explored the nanoenvironment throughout cell cycles and tested different cancer cell lines. We also applied our method to cell samples derived from tissue specimens and fine-needle aspiration (FNA) biopsies in a microfluidic chip. In principle, Cell-TALKING can be easily generalized to explore the nanoenvironments of membrane proteins or TFs. High-throughput sequencing instead of fluorescence imaging may enable us to probe complex nanoenvironments containing tens of biomolecules.

## Results

**Basic principle of Cell-TALKING.** As shown in Fig. 1b, interested DNA modifications on chromatin in single cells are sequentially labeled and crosslinked with respective DNA BPs. A DNA WP is tethered to histone PTM target site via antibody binding. Serving as the molecular tracks in our DNA walking

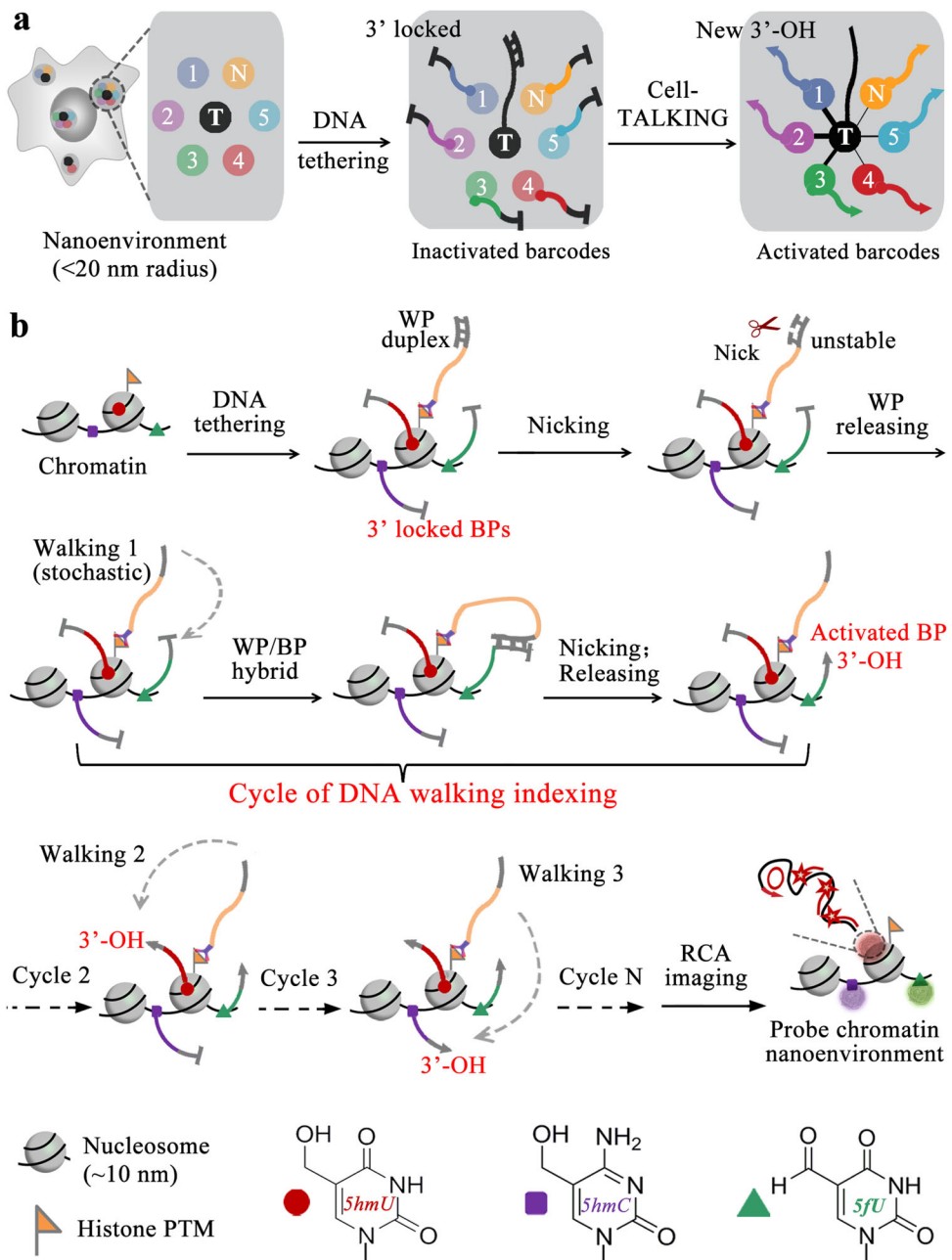

**Fig. 1 The mechanism of Cell-TALKING. a** A conceptual view of probing one-to-many cellular nanoenvironments. Individual DNA probe for T site needs to continuously cleave any proximal probes for interested biomolecules (1, 2, 3,...N) in fixed cells. New 3′-OH ends are generated on the cleaved probes, which will be detected by DNA amplification. **b** Schematic workflow of probing the nanoenvironment of chromatin modifications using Cell-TALKING. The nanoenvironment of three DNA modifications (5hmC, 5hmU, and 5fU) around one histone PTM is detected here as an example.

system, the BPs have unique barcode sequences in the middle and have identical sequences at 3′ end. This 3′ terminal sequence contains several phosphorothioated bases and a base of locked nucleic acid for the inhibition of both 3′ to 5′ digestion and 5′ to 3′ extension, which inactivates the BPs (3′ locked in figures). On the other hand, the WP is sealed at 3′ end with a complementary ssDNA blocker, forming the blocked WP duplex. The 3′ terminal sequence of the WP is also complementary to that of the BPs. A nicking enzyme is utilized to power spatial movement of the WP along adjacent BPs within the nanoenvironments around a PTM site. This enzyme recognizes specific sequence of dsDNA and cleaves only one strand (here the ssDNA blocker or BPs)[37]. During this process, each WP is isothermally released to recognize and cleave any nearby BPs. The cleaved BPs have new 3′-OH

ends, which can be extended by DNA polymerases for isothermal amplification or PCR followed by imaging or sequencing detection. Thus, the 3′ locked BPs are activated. In this work, they are designed to index RCA for fluorescence imaging. In brief, they have a 3′ overhang after the hybridization with DNA circularized probes. Phi 29 DNA polymerase can degrade this 3′ overhang by its inherent 3′ to 5′ proofreading exonuclease activity, and then catalyze RCA reaction using circularized probes as templates. The long ssDNA RCA products (RCPs) enable fluorescence hybridization imaging with single-molecule or single-site resolution, and it allows the detection of low-abundance modifications. Finally, the combination sites of 5hmC/PTM, 5hmU/PTM, and 5fU/PTM within nanoenvironments are indexed with fluorescence bright spots. More complex combinatorial patterns of chromatin

modifications (one-to-two and one-to-many) may be observed or reconstructed by appropriate detection techniques. In contrast, the DNA modifications or their BPs far away from the PTM sites or WPs are not recognized by Cell-TALKING. The uncleaved BPs are 3′ locked and can not induce RCA or other amplification reactions such as PCR. The spatial recognition distance of our method is mainly determined by the sum of the length of BP and that of WP. To explore more modifications or other biomolecules in the nanoenvironments, high-throughput sequencing should be used rather than fluorescence imaging. Notably, the mechanism of the DNA walker used in our method is similar to others[31–36], especially the enzyme-powered ones[31,33–35]. Among them, the design using nicking enzymes[34] is simpler and more robust than those using DNA repair enzymes, exonucleases, or RNase H. In principle, other walkers may also be utilized to perform similar function in our method.

**Cell-TALKING on coverglass and DNA origami**. We first investigated in vitro recognition and indexing performance of Cell-TALKING using fluorescence imaging. We designed different DNA duplex substrates flanking two or three DNA barcoding sequences and individual WP to simply simulate nanoenvironments of biomolecules. Different barcoding sequences indicated the sites of respective biomolecules of interest, and the WP represented the target site. The duplexes were fixed on a coverslip for following DNA walking, amplification, and imaging. After that, the fluorescent RCPs on the coverslip were visualized as bright spots by a laser confocal fluorescence microscopy. Two samples of one-to-two indexing (Fig. 2a) and one sample of one-to-three indexing (Fig. 2b) are tested. As shown in the photographs and intensity curves of arrow lines, the spots were overlapped with two or three fluorescence channels. Related statistical analysis was shown in Supplementary Fig. 1. It indicated the recognition and indexing of multiple adjacent BPs by only one molecule of WP.

We also designed different two-dimensional DNA origami structures to verify TALKING. It is well known that nanoscale distances between different sites on DNA origami are easily controlled and designed. First, we designed three types of BPs surrounding the WP with about 7-nm radius distance as an example. These DNA probes were attached on the origami by DNA hybridizaiton, and the origami was fixed on a coverslip (Fig. 2c). Atomic force microscopy (AFM) was used to confirm our DNA origanmi strucure (Fig. 2c, bottom left). The fluorescence singals of three channels overlapped (Fig. 2c and Supplementary Fig. 2a), which was consistent with our resulsts in Fig. 2b. In constrat, negative controls without probes or with mismatched probes presented very low fluorescence response (Supplementary Fig. 2a), indicating negligible false-positive signals on the origami substrates. Notably, substantial heterogeneity in spot fluorescence overlap was observed. Thus, we quantified the number of spots with fluorescence overlap by five experimental replicates (Supplementary Fig. 2a). A portion of spots missed one or two of all three fluorescence signals, indicating the false-negative level of our method on this origami substrate. This false-negative result may be ascribed to both the limited efficiencies of reactions on coverglass and the failure of the formation of DNA origami substrates. It is well known that molecular reactions such as DNA hybridization and enzymatic reactions are of low efficiency at water/glass interface compared to in solutions. DNA hybridization on DNA origami is also restricted by strong electrostatic repulsion. The yield of DNA origami structures rarely reaches 100%, mainly depending on the complexity of design and experimental conditions. The failure of fluorescence overlap of spots in Figs. 2a and 1b can be explained similarly. Notably, some spots presented relatively large sizes, which may result from

nonuniform RCA efficiency at this interface or the aggregation of multiple copies of DNA substrate molecules.

We further investigated the distance effect on proximity recognition of our method. DNA origami structures were designed to attach one BP with different separation distances (about 7, 14, or 22 nm) to the WP or target site (Fig. 2d). The number of spots was reduced as the increase of separation distances. The normalized spot counts of the samples of other distances to that of 7 nm were shown in the bottom right panel of Fig. 2d. In principle, the maximum recognition distance could reach the sum of the length of BP and that of WP. However, the stretching of flexible DNA may be restricted at water/glass interface or by strong electrostatic repulsion from DNA origami. Increasing the length of probes may extend the recognition distance. Here we used longer WPs containing an additional spacer (15 or 30 nt) to investigate this issue. As shown in Supplementary Fig. 2b, longer WPs increased the signal response at long-distance detection sites. This result confirmed the probe length-determined recognition range of our method. Therefore, our method achieved one-to-many proximity recognition and indexing performance. These results may enable the exploration of combinatorial patterns of diverse chromatin modifications in the same nanoenvironments.

**Bioorthogonal labeling of DNA modifications**. We have previously reported the differentiated labeling of 5hmU and 5hmC[20]. The 5hmU sites in dsDNA were first labeled with a phosphate or thiophosphate group by 5-hydroxymethyluridine DNA kinase (5-HMUDK), and then the 5hmC was labeled with an azido group by T4 Phage $\beta$-glucosyltransferase (T4 $\beta$-GT). 5-HMUDK can catalyze the transfering of the $\gamma$-phosphate of adenosine triphosphate (ATP) to the hydroxymethyl moiety of 5hmU in dsDNA. Nevertheless, abundant natural phosphate groups or thiol groups exist in cells, and the bioorthogonal labeling of 5hmU by 5-HMUDK remains a challenge. An ATP analog containing bioorthogonal crosslinking groups is required. We here synthesized ATP-$\gamma$-alkyne, an ATP analog with modified $\gamma$-phosphate, based on a synthetic methodology reported by previous papers[38,39]. Propargylamine was used to synthesize ATP-$\gamma$-alkyne with the help of 1-ethyl-3-(3-dimethyllaminopropyl) carbodiimide hydrochloride (EDC-HCl). The product after purificaiton was analyzed by mass spectrometry (Fig. 3a). Its structural characterization was also shown in Supplementary Fig. 3 and Supplementary Materials. This bioorthogonal ATP analog was then used to label 5hmU in dsDNA samples by 5-HMUDK. The dsDNA sample containing 5hmU:A sites within the recognition sequence for NcoI endonuclease was prepared. This dsDNA substrate was treated with 5-HMUDK for phosphorylation reaction. After that, the reaction product was digested by NcoI to cut dsDNA with 5hmU:A rather than phosphorylated 5hmU:A in the recognition sequence. The DNA melting curves in Fig. 3b showed that the cut of this reaction product by NcoI was blocked. Therefore, we demonstrated that ATP-$\gamma$-alkyne was an efficient phosphate donor for 5-HMUDK to label 5hmU. Furthermore, we investigated the reduction of 5fU to 5hmU. Sodium borohydride is a small water-soluble reductant and has been used to reduce 5fC to 5hmC in genomic DNA[40]. This reductant was also shown to convert 5fU to 5hmU (Fig. 3c). Thus we can label 5fU by the method of 5-HMUDK-catalyzed 5hmU labeling after reduction. These results may enable the labeling and barcoding of 5hmU, 5hmC, and 5fU in cells following a proper order.

**Cell-TALKING exploring nanoenvironments of chromatin modifications**. We first demonstrated the feasibility and specificity of Cell-TALKING in fixed cells. The secondary antibody for

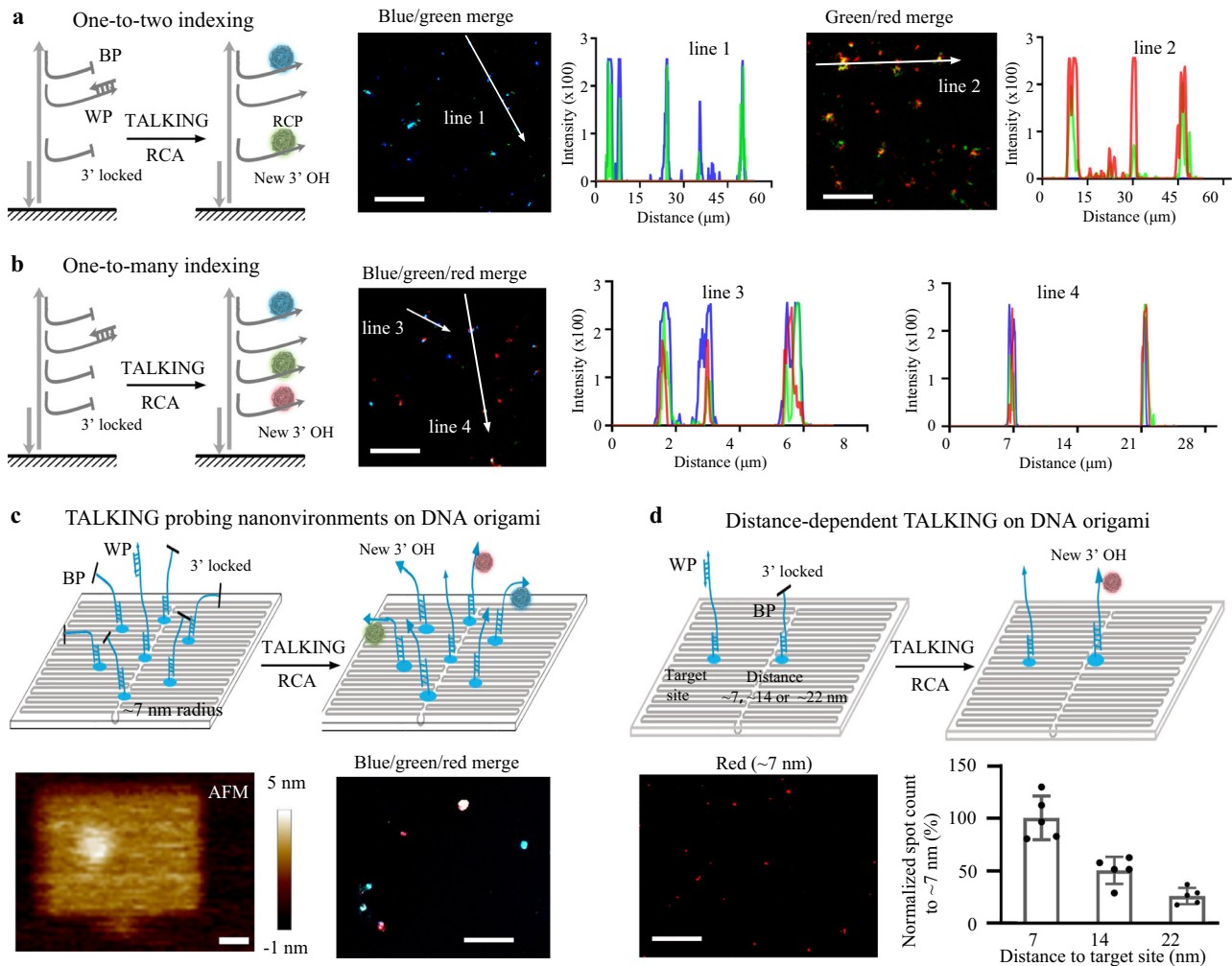

**Fig. 2 Proof-of-principle experiments of in vitro Cell-TALKING. a** The one-to-two detection performance on DNA duplex substrates. **b** The one-to-three detection performance on DNA duplex substrates. **c** The one-to-many detection performance on DNA origami substrates. In **a**–**c**, five times of the experiment were repeated independently with similar results. **d** The distance-dependent detection performance on DNA origami substrates. The DNA duplex or DNA origami is modified on the surface of a coverglass. The scale bars of images are 25 and 20 μm in Fig. 1a, 10 μm in Fig. 1b, 20 nm and 5 μm in Fig. 1c, 15 μm in Fig. 1d, respectively. The BP and WP indicate the barcoding probe and walking probe, respectively. Five repeated experiments were performed for statistical analysis of spot count of randomly selected images, and the data are presented as mean values ± SD.

the histone PTM was crosslinked with an oligonucleotides for the hybridization of the WP. It has been previously shown that one-to-four DNA molecules were conjugated per antibody molecule and the major conjugated product contains only one DNA molecule[41]. We used ATP-γ-alkyne to realize the bioorthogonal labeling of intracellular 5hmU and/or 5fU with respective DNA primer probes for RCA and hybridization imaging (Supplementary Fig. 4). Almost all sites of 5hmU were crosslinked with DNA probes for 5hmU assay, and no residual sites were detected by the DNA probes for 5fU assay (Supplementary Fig. 4b). These results enabled successive labeling of 5hmU and 5fU in the same cells (Supplementary Fig. 4c). Therefore, we would follow the labeling order of 5hmU, 5hmC, and 5fU to achieve the discrimination of these three DNA modifications. The cellular specificity of Cell-TALKING was also confirmed by several negative control experiments. These negative controls mainly include the ones without certain vital elements such as regconition enzymes (5-HMUDK or T4 β-GT), antibodies or nicking enzyme, and the ones with the mismatched DNA probes. They all induced nearly zero or very low background fluoresence signals (Supplementary Fig. 5), which indicated the negligible nonspecific signal accumulated by our Cell-TALKING method.

Then the nanoenvironments around H3K27ac, H3K27me3, H3K4me3, H3K4me1, or γH2AX were investigated, respectively. It is well known that H3K27ac, H3K4me3, and H3K4me1 are enriched in active chromatin or euchromatin, and H3K27me3 is regarded as a repressive mark in heterochromatin. In addition, γH2AX plays an important role in DNA double-stranded break and repair. It is mainly enriched in euchromatin at the early stage of DNA damage response and distributed in heterochromatin during later times[42]. Figure 4a showed selected merged images of cells for these five histone PTMs. The organization of DNA modifications around these active or repressive PTMs differed from each other. The red, green, and blue spots in the images indicated the combination sites of 5hmU/PTM, 5fU/PTM, and 5hmC/PTM, respectively. Both fluorescence intensity and bright spot count in single cells of each channel were extracted to indicate molecular information of complex combinations of chromatin modifications (Supplementary Fig. 6). The averaged spot percentages of three combination sites were shown in Fig. 4b. This percentage means the proportion of the spot count of one combination modification in that of all three in the same single cells. Notably, some spots presented relatively large sizes. They may

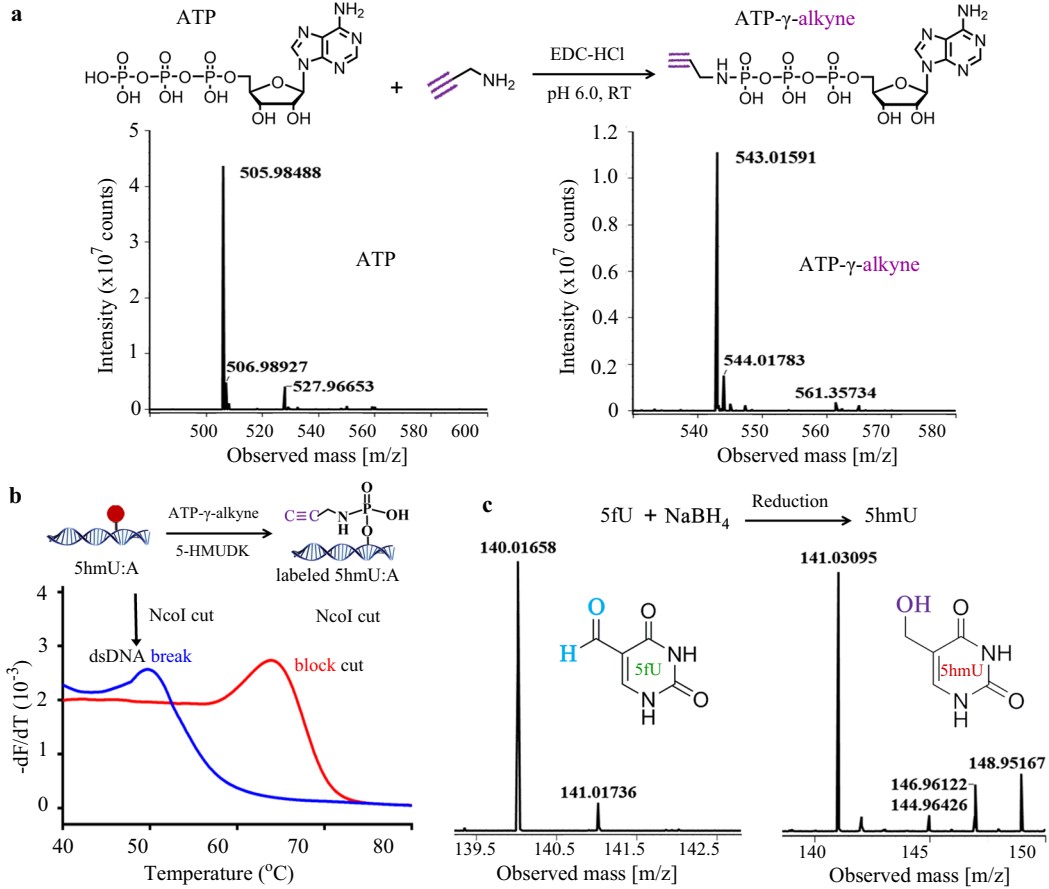

**Fig. 3 Synthesis of ATP-γ-alkyne and labeling of DNA modifications. a** MS characterization of ATP-γ-alkyne synthesis. Left panel, the mass spectrum of ATP ($C_{10}H_{15}N_5O_{13}P_3$, [M + H]⁻: 505.98847, calculated; 505.98488, found); right panel, the mass spectrum of ATP-γ-alkyne ($C_{13}H_{18}N_6O_{12}P_3$, [M + H]⁻: 543.02010, calculated; 543.01591, found). **b** Melting curve analysis of the cleavage products of 5hmU-contained dsDNA after labeled by 5-HMUDK and ATP-γ-alkyne. **c** MS characterization of 5fU reduction. Left panel, the mass spectrum of 5fU ($C_5H_4N_2O_3$, [M + H]⁻: 140.02274, calculated; 140.01658, found); right panel, the mass spectrum of reduction product 5hmU ($C_5H_5N_2O_3$, [M + H]⁻: 141.03057, calculated; 141.03095, found).

suggest the adjacency of multiple combination sites or result from nonuniform RCA efficiency in the molecularly crowded cell nucleus[21]. Thus a fluorescence spot may consist of one or more combination sites. The spot numbers or signal intensities of single cells may reflect not only the levels of modifications but also their accessibility to the molecular probes. Only the exposed or accessible modification sites were easily labeled and detected. We also used PLA as a a complementary method to confirm the efficacy of our method (Supplementary Fig. 7). Moreover, most of the spots of different combinations were individually distributed throughout the nuclear volume. But a few spots appeared to be spatially overlapped via fluorescence imaging (250–400-nm optical resolution). The overlap of different fluorescence spots may indicate different DNA modifications that are within the same nanoenvironment. As an example, we selected two projected images and relevant three-dimensional analysis of multiple z-stack images (Fig. 4c and Supplementary Fig. 8). To conceptually investigate this issue, here we roughly treated the overlap within large distances (about hundreds of nanometers) as hypothetical spatial colocalization of different combinations of chromatin modifications. We observed the hypothetical one-to-two site and one-to-three site by three-dimensional analysis. Especially, strict and systematic experiments are required to verify above results, and it may be further determined by super-resolution fluorescence imaging or nonoptical detection techniques such as DNA sequencing (see "Discussion" below).

**Dynamic organization of nanoenvironments of chromatin modifications during cell cycles**. To investigate the changes of nanoenvironments of chromatin modifications throughout the cell cycle, we probed H3K4me3 and H3K27ac as two examples. The MCF-10A cells were arrested in G1 phase, S phase, and G2 phase, respectively. Selected merged images of cells were shown in Fig. 5a. We summarized the signal intensities of those three combinational chromatin modifications in single cells (Supplementary Fig. 9a), and showed the averaged spot counts of single cells in Fig. 5b. Most averaged spot counts increased from G1 to S phase, which may reveal the generation of these combination sites along with DNA or chromatin replication. The 5fU/H3K4me3 combination sites are an exception, implying a different generation mechanism or more complex dynamics. We further calculated the spot percentages of three combination patterns (Fig. 5c and Supplementary Fig. 9b), and found that the percentage of 5hmU/PTM was much higher than other two combinations throughout these cycle phases. All these results demonstrated that these combinations of chromatin modifications were dynamically organized during the cell cycle. Nevertheless, systematic experiments are required to uncover their biological functions in chromatin replication and cell division.

**Exploring nanoenvironments of chromatin modifications in cancer cell lines and clinical specimens**. Next, we characterized the nanoenvironments of chromatin modifications in different

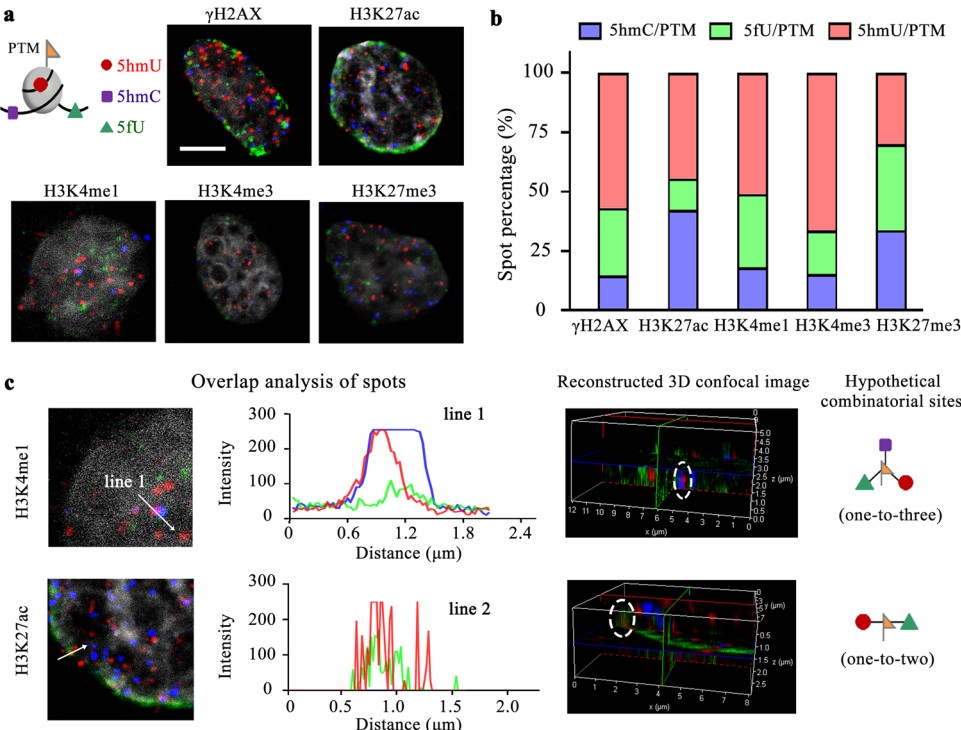

**Fig. 4 Exploring different nanoenvironments of chromatin modifications in single cells. a** Selected merged cell images for five histone PTMs. The red, green, and blue fluorescence spots each indicated the combination sites of 5hmU/PTM, 5fU/PTM, and 5hmC/PTM. The scale bar is 5 μm. **b** The spot percentages of three combination sites of single cells in five samples (cell numbers = 52, 34, 32, 43, and 38, respectively). **c** Analysis of the overlap or cluster of different spots. For the experiments of cell imaging in all figures, five times were repeated independently with similar results.

cancer cell lines. Three human breast cell lines, including non-tumorigenic MCF-10A, carcinogenic MCF-7, and highly invasive MDA-MB-231 were used as examples. We detected three DNA modifications around H3K4me3, and the merged images of cells were shown in Fig. 6a. These cell lines had different levels of three modification combinations according to their single-cell spot count and fluorescence intensity (Supplementary Fig. 10). The averaged value of 5hmU/H3K4me3 combination in MDA-MB-231 is lower than that in MCF-10A or MCF-7, possibly implying the potential negative effect of 5hmU/H3K4me3 combination on cancer metastasis. Other two combinations do not present remarkable changes in these three cell lines. Furthermore, the distances of each spot to the nucleus periphery were also calculated by MATLAB (Fig. 6b). We observed that the distances of all three combinations in MCF-10A are lower than those in MCF-7 and MDA-MB-231. It suggests that such distances of these modification combinations might be related to cancer progression and metastasis.

Moreover, we applied Cell-TALKING to clinical specimens with the integration of a microfluidic chip (Fig. 6c). Microfluidic techniques have been widely used to capture, culture, or/and fix rare cells. We designed a simple chip as an example for following experiments mainly including cell culture, fixation, reaction, and imaging. Cell suspensions derived from tissues or FNA biopsies of lung cancer patients were tested. We still detected the nanoenvironments containing 5hmU, 5fU, and 5hmC around H3K4me3. As shown in Fig. 6c, averaged spot counts of 5hmU/H3K4me3 combination were higher than those of 5hmC/H3K4me3 or 5fU/H3K4me3 in these cancer samples. It may imply potential positive function of 5hmU/H3K4me3 combination in cancer. In contrast, quite low levels of 5hmC/H3K4me3 combination in most cancer samples may suggest its potential negative correlation with cancer. Despite these quite simple exploration, systematic experiments are required to validate those superficial speculations and to

further clarify the relevance of chromatin modification nanoenvironments to cancer or other diseases[20,43]. In principle, Cell-TALKING may also be used to analyze tissue sections with routine manipulation processes. Detecting section samples may identify cell subtypes and uncover the spatial relationship between different single cells. In addition, our method can be applied to other samples such as embryo stem cells or nerve cells to explore the function of combinational chromatin modifications in differentiation and development.

## Discussion

In this work, we have demonstrated that Cell-TALKING can probe cellular nanoenvironments containing diverse biomolecules. In the context of chromatin, we have explored the nanoenvironments containing three DNA modifications around histone PTMs. Cell-TALKING can index different relationships of diverse chromatin modifications, and DNA barcoding amplification is used to improve fluorescence hybridization imaging with single-site or single-molecule resolution. We have identified diverse combinatorial chromatin modifications, and obtained their molecular information about site numbers and spatial distribution. We have investigated their dynamic changes during cell-cycle stages, and explored their molecular signatures of several cancer cell types. We successfully applied this method to cell suspensions from clinical specimens with microfluidic techniques. Our method helps uncover the nanoscale composition and organization of chromatin modifications and may deepen the understanding of their function in physiological and pathological processes.

Notably, both BP and WP used in this DNA walking indexing system are tethered to the macromolecules of chromatin. The sum of the length of these probes roughly determines the maximum distance of molecular recognition. It may allow to detect long range distribution or contact of chromatin components by

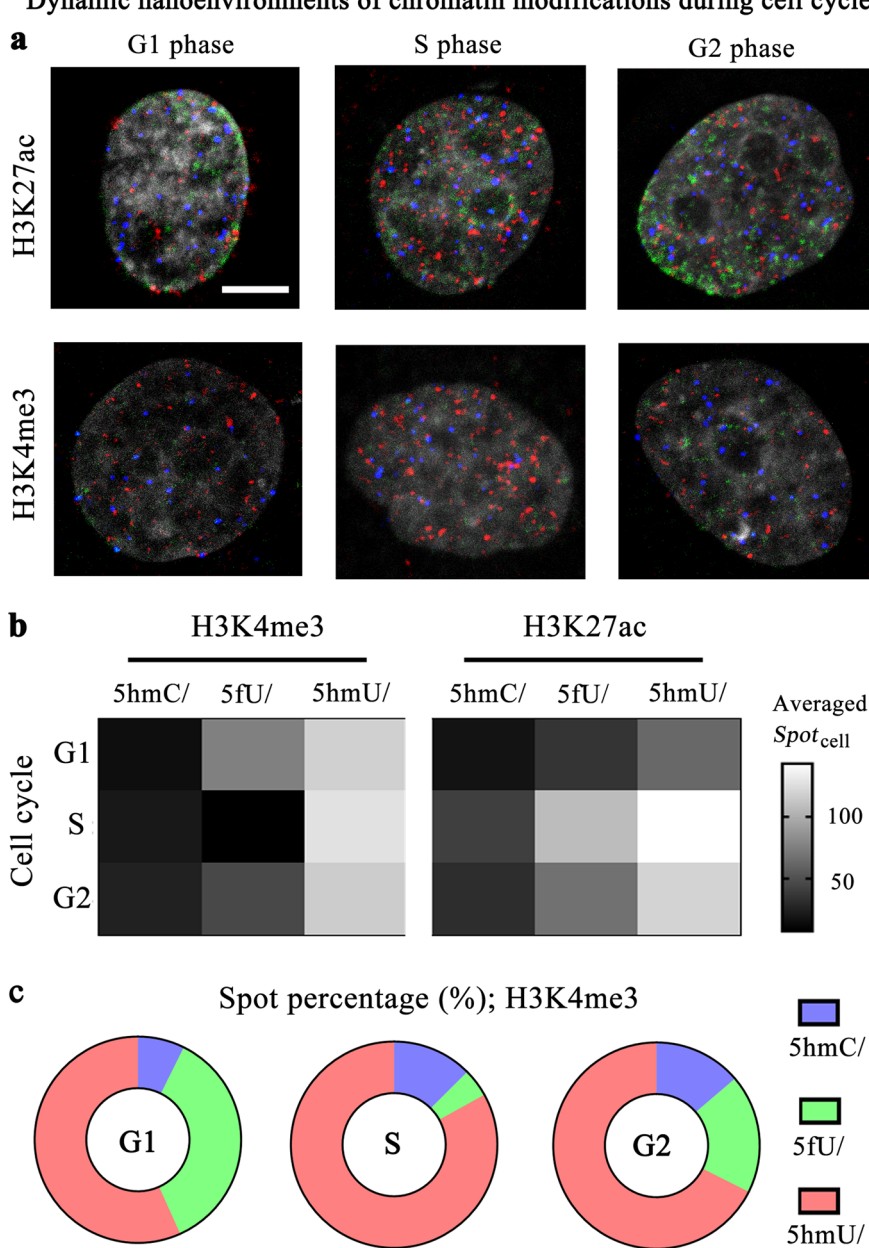

**Fig. 5 Dynamic changes of nanoenvironments of chromatin modifications during cell cycles. a** Selected merged cell images for three phases of cell cycle. The scale bar is 5 μm. **b** The changes of the averaged single-cell spot counts of three combinational chromatin modifications. The cell numbers of G1, S, and G2 phases are each 45, 43, and 52 for H3K4me3, and 38, 37, and 50 for H3K27ac. **c** The spot percentages of three combination sites of single cells.

increasing the length of these probes. Besides chromatin modifications, chromatin-associated proteins such as TFs or remodeling proteins are important to regulate chromatin structure and function. In principle, our method is easily generalized to probe the nanoenvironments around the target sites of these proteins, such as spatial organization of multiple TFs or between TFs and chromatin modifications. Routine fluorescence imaging can only simultaneously detect three or four targets with limited orthogonal fluorophores. Recently, sequential DNA hybridization-image-erase techniques have been developed to profile tens of to hundreds of RNA or protein species in single cells[43–45]. Combined with these techniques, our method may be exploited to visualize more complex organization of biomolecules within cellular nanoenvironments.

As described in the introduction, the analysis of cellular nanoenvironments is quite different to both the detection of two

proximity targets by PLA and even the independent, parallel PLA analysis. Very recently, we proposed a one-to-one pairwise proximity-differentiated DNA recognition mechanism based on the skillfully integrated design of an improved DNA proximity ligation and toehold-mediated strand displacement[46]. We have used this mechanism to investigate the spatial proximity of two DNA modifications (5fC and 5hmC). It is able to successively encode 5fC/5hmC pairwise proximity sites, residual 5fC sites, and residual 5hmC sites with respective circularized DNA barcodes as RCA templates. It overcomes the limitation of current PLA methods that they can not detect the residual sites of two analytes. However, this pairwise proximity-differentiated mechanism can not probe one-to-many proximity within nanoenvironments because only a single proximity record is generated from each DNA probe. Even its multiplex version can only detect multiple one-to-one proximities, but these proximities are independent

**a** Cell type-dependent nanoenvironments around H3K4me3

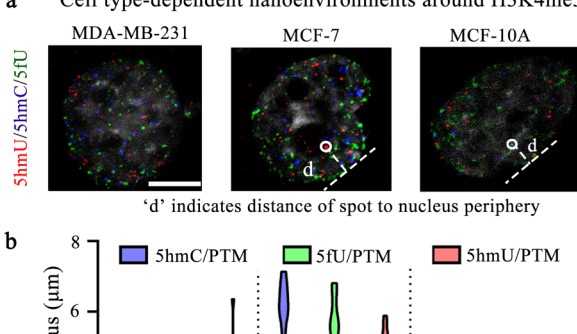

'd' indicates distance of spot to nucleus periphery

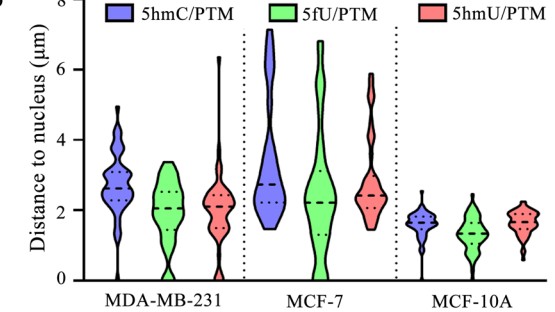

**c** The nanoenvironments around H3K4me3 of clinical specimens

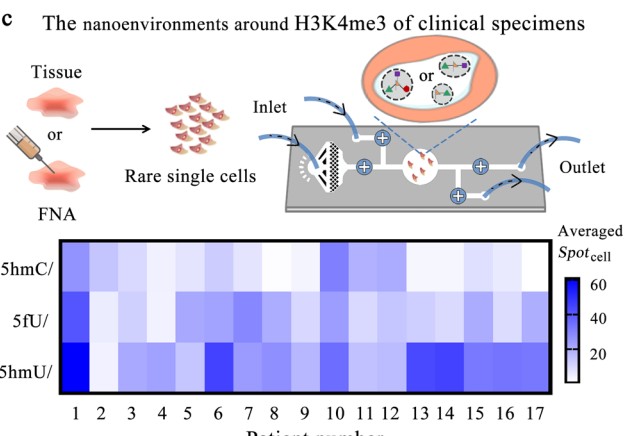

**Fig. 6 Exploring nanoenvironments of chromatin modifications in cancer samples. a** Selected merged cell images for three breast cell lines. The letter "d" indicates the distance of the spot to nucleus periphery. The scale bar is 5 μm. **b** Statistical analysis of the distances of spot to the nucleus periphery. The cell numbers are each 52, 44, and 64. **c** The detection of clinical tissues (No. 1–14) and FNA samples (No. 15–17) with a microfluidic chip. Upper panel, the schematic illustration of the workflow; bottom panel, heat map showing the averaged single-cell spot counts of three combinational chromatin modifications.

pairs rather than the one-to-many nanoenvironments. In order to overcome this challenge, we here design our Cell-TALKING method based on DNA proximity walking indexing. But the Cell-TALKING method is unable to detect the analyte sites outside the nanoenvironments. A modified Cell-TALKING version combined with our aforementioned proximity-differentiated concept may be feasible to differentially detect the sites within and outside the nanoenvironments in the same cells. Furthermore, PTMs recognition via antibodies used in our method may be affected by epitope occlusion[47]. The specific binding of antibodies to a single histone PTM could be obfuscated by any neighboring PTMs at the same tail. This steric occlusion effect is a common issue for PTMs recognition, especially on highly modified histones. It requires to generate more specific antibodies against histones containing various modification valences[48]. On the other hand, the detected combinatorial chromatin modifications by our method may locate on different genomic regions and even on different chromatins ascribed to complicated chromatin

interaction in cell nucleus. Due to the large size of antibodies (about 150 kDa, ~15 nm), antibodies-used methods may not accurately determine whether two coexisting PTMs locate on the same histone tail or not. The separation distance of two PTMs on the same histone tail is about 4 nm[49]. Genetically encoded multivalent protein sensors with small sizes have been reported to selectively detect combinatorial PTMs on the same histone tail[49]. In addition, the emerging nanobodies with the size of about 4 nm (12–30 kDa) may be also promising tools[50–52].

In addition, other issues of our work should be underlined. The Cell-TALKING mechanism can repeatedly produce cleavage records of any nearby BPs by the cycling of WPs. It enables "one-to-many" molecular indexing in the nanoenvironments and overcomes the limitation of current "one-to-one" PLA techniques that record only a single combination of each probe. New 3′-OH ends are generated for DNA extension and amplification, amplicons of which can be detected by fluorescence imaging, DNA sequencing, or other downstream detection techniques. Despite the molecular recognition resolution of <20 nm, fluorescence imaging with a confocal laser fluorescence microscope is used to detect the barcoding amplicons in this work. This microscope as detection tool is restricted to low optical resolution (diffraction limit levels, 250–400 nm). It is disabled to precisely determine whether several spots with high fluorescence colocalization or overlap rates are localized at the same nanoenvironments. Newer, specialized super-resolution microscopies have been developed to improve the lateral optical resolution to 20–100 nm. In principle, they can directly estimate the proximity of biomoleculs by fluorescence colocalization without biochemical recognition reactions such as DNA proximity. However, this resolution can vary considerably in optically complex samples, and the z-axis resolution is still limited[53]. Furthermore, registration errors from sample drift especially in multichannel imaging often lead to incorrect colocalization analysis[54]. Nevertheless, conbiming Cell-TALKING or its improved versions with super-resolution imaging may allow more accurate detection of the composition and distribution of diverse molecules in the same nanoenvironments.

Recently, microscopy-by-sequencing as an emerging technique, has been established to probe the spatial organization of biomolecules inside cells[55,56]. It can avoid the optical resolution limitation and provide a global spatial map for a complex molecular population by using DNA proximity recognition and high-throughput sequencing. Inspired by these advancements, our Cell-TALKING method may be intergrated with sequencing for the profiling of accurate nanoscale organization of even tens of or hundreds of components. Especially, during the revision of our manuscript, one method directed at the same goal, Nano-Deep, was published[57]. NanoDeep used pre-assembled DNA nanoassemblies with multiple position barcodes to translate spatial organization information of membrane proteins into DNA sequencing readouts. This method can probe the nanoenvironments of one-to-many membrane proteins (three or four other proteins surrounding Her2 protein), but it required hundreds of thousands of cells as start materials. Its mechanism of DNA proximity recognition and barcoding is quite different to that of our Cell-TALKING method. Notably, the pre-assembled DNA nanoassemblies own a size of about 34 nm. This relatively large size may hinder its application in molecularly crowded intracellular environments especially cell nucleus. And single-cell sequencing analysis of cellular nanoenvironments may be still a challenge. Another issue is that Cell-TALKING can not record the "many-to-many" molecular combinations or relationships. It is highly desirable to probe any or a complete set of combinatorial patterns between groups of molecules in the same nanoenvironments. Conceptually, it requires to record nondestructive and continuous interaction of any two adjacent DNA probes.

## Methods

**Materials**. Adenosine 5′-triphosphate (ATP) disodium salt, propargylamine, EDC-HCl, (3-mercaptopropyl)trimethoxysilane, and Dibenzocyclooctyne-sulfo-N-hydroxysuccinimidyl ester were purchased from Sigma Aldrich (St Louis, USA). Sodium borohydride, copper(II) sulfate, sodium ascorbate, and psoralen were obtained from Aladdin (Shanghai, China). UDP-6-azide-glucose (UDP-N$_3$-Glu) were obtained from Jena Bioscience (Jena, Germany). The reactions were performed in RNase-free water and buffers. All chemicals were used as received without further purification. The oligonucleotides used in this work (Supplementary Table 1 and Supplementary Data 1) were synthesized by Sangon Biological Co. Ltd (Shanghai, China). dhmUTP was purchased from Trilink Biotechnologies (California, USA). T4 DNA ligase, Exonuclease I, DNase I, and Exonuclease III were purchased from Takara Biotechnology Co. Ltd (Dalian, China). M13mp18 single-stranded DNA, Nt.BbvCI, Klenow Fragment (3′ → 5′ exo-), hSMUG1, shrimp alkaline phosphatase, 5-HMUDK, phi29 DNA polymerase, and T4 β-GT were purchased from New England Biolabs Ltd. (Beijing, China). H3K4me1, H3K4me3, H3K27me3, γH2AX, and H3K27ac were purchased from Cell Signaling Technology (Massachusetts, USA). Donkey anti-Rabbit IgG (H + L) Highly Cross-Adsorbed Secondary Antibody, and nuclease S1 were obtained from ThermoFisher Scientific. QuickBlock™ blocking buffer for immunostaining, QuickBlock™ primary antibody dilution buffer for immunostaining and QuickBlock™ secondary antibody dilution buffer for immunofluorescence were all obtained from Beyotime Biotechnology (Shanghai, China).

**Preparation of circularized probes**. The circularized probes were prepared by a hybridization, ligation, and digestion multistep reaction. The hybridization reaction was carried out in 10 μL 1× T4 DNA ligase reaction buffer containing 10-μM padlock probes and 30-μM ligation linker at 55 °C for 2 h. Then, 3-μL T4 DNA ligase (350 U/μL) was added to catalyze the ligation at 37 °C for 2 h, followed by inactivation at 65 °C for 30 min. Finally, excess Exonuclease I and Exonuclease III were added to digest linear DNA. The mixture was heated at 80 °C for 30 min to inactivate enzymes. The resulting circularized probes were stored at −20 °C.

**One-to-many molecular recognition and indexing on coverglass by Cell-TALKING**. The DNA duplex substrates were assembled by mixing 5-μM glass-capture substrate, 6-μM glass-WP (previously blocked by walking blocker probe), and 6-μM glass-BP into 1x NEBuffer 2. The mixture was incubated at 95 °C for 15 min, 55 °C for 4 h, and 45 °C for 5 h, respectively. The reaction products were stored at 4 °C until the next reaction.

The coverglass was treated with a mixture of concentrated H$_2$SO$_4$ and 30% H$_2$O$_2$ (3:1 v/v) for 1 h and then immersed in the solution of (3-mercaptopropyl) trimethoxysilane (2%, v/v) for surface thiosilanization. Self-made polydimethylsiloxane (PDMS) cavity (φ 4 mm) was adhered to the silanized coverglass using a double-sided adhesive tape (120-micron thick, Grace BioLabs), which formed the reaction chambers. Then, 3-μM glass-modified primer was added into the reaction chambers for crosslinking reaction (45 °C, 3 h). After washing three times with 1×PBS buffer, the primer-functionalized coverglass was used to capture the DNA duplex substrates by hybridization reaction.

For the reactions of Cell-TALKING on coverglass, 20-μL 1× Cutsmart buffer containing 5 U Nt.BbvCI was added in the reaction chamber and then incubated at 37 °C for 2 h. After washing three times with 1×PBS buffer, circularized padlock mixtures (200 nM each) in 2x SSC hybridization buffer were added for the hybridization reaction at 37 °C for 2 h. After washing, 10-μL 1x phi29 DNA polymerase buffer containing 5 U phi29 DNA polymerase and 2.5-mM dNTPs was added for RCA reation (37 °C for 2 h). Finally, fluorophore-labeled DNA probes were captured by RCPs in the reaction chamber for imaging analysis.

**Probing the nanonvironments on DNA origami by Cell-TALKING**. Design and assembly of two-dimensional DNA origami structures. The detailed sequences of DNA origami structures were shown in Supplementary Data 1. The DNA origami was prepared by a long scaffold strand (M13mp18 single-stranded DNA) and hundreds of short staple strands. In brief, 5-nM scaffold strand and 50-nM respective staples in 1x TAE/Mg$^{2+}$ buffer (40 mM Tris, 2-mM EDTA, and 12.5-mM magnesium acetate, pH 7.0) were annealed following the procedure: 95 °C for 5 min, 95 °C to 25 °C at a rate of 0.1 °C/10 s. Then, 100-nM biotin probe was added and the mixture was incubated at room temperature for 30 min. Then the biotin-functionalized DNA origami was purified by Amicon Ultra-0.5 NMWL 100-kDa centrifugal filter (Merck Millipore).

Characterization of DNA origami by AFM. AFM tips (model PeakForce fluid; Bruker multimode 8) were used in this work. First, a freshly cleaved mica surface was prepared for sample mounting. Three microliter of DNA origami (1 nM) was pipetted onto the center of the mica surface for 3-min incubation. The samples were imaged in tapping mode in solution using DNP-10 tips (Bruker's robust Silicon Nitride AFM probe).

Cell-TALKING on DNA origami. It is well known that DNA extension reaction by polymerase can destroy DNA origami, which contains lots of 3′-OH in duplexes. Thus, we used psoralen to covalently crosslink DNA duplexes of DNA origami, which resists the polymerase-catalyzed extension reaction. In brief, 1-mM psoralen solution was used under 365-nm irradiation for 1 h on ice. Then the crosslinked

DNA origami containing biotin probes (mentioned above) was captured by streptavidin-modified coverglass. Subsequently, respective glass-BP (5 nM) and blocked glass-walking pr1.obe (5 nM) were captured on this DNA origami to form a nanonvironment (~7-nm radius) or a distance-dependent sample. The reactions of Cell-TALKING on DNA origami are similar to those on coverglass mentioned above.

**Chemical synthesis of ATP-γ-alkyne**. The detail is shown in the supplementary information. In brief, ATP and propargylamine were used to synthesize ATP-γ-alkyne in the prescence of EDC-HCl. Light-yellow viscous products were obtained and then were purified with silica gel column chromatography to get the white solid ATP-γ-alkyne.

**5fU reduction to 5hmU**. Sodium borohydride solution (3.5 μg/mL) was freshly prepared and added into the solution of 5fU nucleobases (10 μg/mL) slowly. The vortex reaction was carried out at room temperature in the dark for 1 h. After that, sodium acetate solution (750 mM, pH 5) was added slowly to stop the reaction at room temperature until no gas was released. 5fU: HRMS (ESI, negative mode) for C$_5$H$_4$N$_2$O$_3$, [M + H]$^-$: 140.02274 (calculated), 140.01658 (found). 5hmU: HRMS (ESI, negative mode) for C$_5$H$_5$N$_2$O$_3$, [M + H]$^-$: 141.03057 (calculated), 141.03095 (found).

**Preparation of 5hmU-contained DNA duplex**. The model dsDNA substrates containing the site of 5hmU:A were prepared by in vitro DNA polymerase-catalyzed extension of a primer-template duplex. For the reactions of 5hmU phosphorylation by 5-HMUDK, 10-μM dsDNA substrate was incabated with 20 U 5-HMUDK and 1-mM ATP-γ-alkyne in 1× Cutsmart buffer at 37 °C for 1 h. Then the DNA products were cut by 10 U NcoI-HF in 1x Cutsmart buffer at 37 °C for 1 h, followed by melting curve analysis (LightCycler 96, Roche).

**Cell culture and clinical samples**. MCF-10A, MCF-7, and MDA-MB-231 cells were cultured in Dulbecco's modified Eagle's medium with 10% fetal-bovine serum and 1% antibiotics penicillin-streptomycin (100 U/mL) in a humidified incubator containing CO$_2$ (5%) at 37 °C. Lovastatin, hydroxyurea, and colchicine were used to synchronize cells in G1, S, and G2 phase, respectively. The study of human tissue samples and FNA biopsies was approved by ethics committee of Xi'an Jiaotong University (No. 2020-216) in accordance with the law on human experiments. Informed consent was obtained from all patients. These clinical tissue and FNA samples were treated with collagenase to collect cell suspensions, and then these cell suspensions were analyzed in the microfluidic chip.

**Cell-TALKING probing nanoenvironments of chromatin modifications**. The step-by-step protocol is shown in the supplementary information. For a typical experiment, the cells were successively treated with cell fixation, the chemical labeling of different DNA modifications, the binding of histone PTM with anti-bodies, DNA proximity nicking/walking reaction, the hybridization of circularized padlocks, 3′ to 5′ digestion by Phi 29 DNA polymerase, RCA, and the hybridization of fluorophore-labeled DNA probes. The nuclei were stained using DAPI. Cells were washed three times before fluorescence imaging.

**Design and fabrication of microfluidic chip**. The microfluidic chip was designed by a computer-assisted design software and the cast mold (50-μm thick) was prepared by photolithography using AZ 50XT positive photoresist on a polished silicon wafer. The chip was controlled by torque-activated valves, consisting of two layers of PDMS (oligomer and cross-linker at a ratio of 10:1 (w/w)). In brief, a 100-μm-thick PDMS layer was spin-coated onto the cast mold to form the channels of the chip. After curing at 70 °C for 30 min, the hex nuts and nylon screws were embedded in another uncured PDMS layer over corresponding positions of the channels, then followed by the final curing step. The separate inlets and outlets, combining with torque-activated valves, function as two independent in and out systems for clinical samples and reagents. The chamber supports cell culture, subsequent fixation, and Cell-TALKING reactions, also provides the flexible place for cofocal imaging.

**Imaging and data analysis**. Fluorescence imaging was performed on a laser scanning confocal microscopy (Leica) with a 63× water objective (NA 1.2). Fluorescence images were acquired as z-stacks at intervals of 0.5 μm. The z-stacks of images were combined to a single one by maximum intensity projection using LAS X Version. Cells were randomly selected. Matlab 2019b was used to extract the spot fluorescence intensity, spot count, and the distance of spot to the nucleus periphery of individual cells.

**Reporting summary**. Further information on research design is available in the Nature Research Reporting Summary linked to this article.

## Data availability
The data that support the findings of this study are available from the corresponding author on reasonable request. Source data underlying Figs. 2d, 4b, c, 5c, and 6b are available as a Source data file provided with this paper.

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

## Acknowledgements

This research was supported by the National Natural Science Foundation of China (Grant number 92068118, 31671013, 21705124, and 21874105), the China Postdoctoral Science Foundation (Grant number 2017M613102, 2018T111032, and 2019M663658), the Natural Science Basic Research Program of Shaanxi (Grant number 2018JC-001, 2020JQ-020, and 2020JQ-021), the Fundamental Research Funds for the Central Universities and "Young Talent Support Plan" of Xi'an Jiaotong University.

## Author contributions

Y.X.Z. directed the research. F.C., M.B., and Y.X.Z. conceived and designed the experiments. F.C., M.B., X.C., J.X., Y.Z., and N.W. carried out experiments and performed data analysis. L.W. and D.X.Z. provided clinical samples and performed the clinical result analysis. F.C., M.B., and Y.X.Z wrote and revised the manuscript.

## Competing interests

The authors declare no competing interests.
