## [Peer Review File · Nature Communications]

REVIEWER COMMENTS

Reviewer #1 (Remarks to the Author):

In this manuscript, the authors reported a newly developed method (Cell-TALKING) to explore nanoenvironments of chromatin modifications, which was applied in breast cancer cell lines and clinical specimens. This methodology can be used for analyzing single-cell chromatin nanoenvironments, where the modification sites in the nanoenvironment are labeled with respective DNA barcodes by Cell-TALKING, and then visualized by rolling circle amplification (RCA)-improved fluorescence imaging. The method itself is intriguing, however, there are several major issues:

1. The method relies on many sequential steps and it is currently unclear what is the level of non-specific signal that accumulates? There are several controls that are missing in order to understand the expected signal/noise ratio?
2. The method relies on antibodies, similarly to ChIP-seq. Epitope occlusion is a major issue in antibody recognition in these techniques and I have not seen controls to address the issue or any discussion of it in the manuscript.
3. As the authors mention, they have recently published several similar techniques probing epigenetic modifications in single cell. It seems that the major advancement in this paper is the capacity for multiplexing, but it is not clearly indicated. If the leap is in multiplexing, the authors should provide new epigenetic insights or novel biology to justify publication in Nature Communications.
4. The physiopathological relevance of the chromatin nanoenvironment reported by the authors and the role it plays in cancer development remain unclear and should be better discussed.
5. While the method provides information of proximity, there is still no validation that marks that are detected to be close in space by this method are in fact, for example, on the same tail and represent an epigenetic state.
6. For a methodology paper, the exact methodological process and protocol are not clear or detailed enough and an elaborated materials and methods should be included in the supplementary information.

Reviewer #2 (Remarks to the Author):

The manuscript reports a new method for the multiplexed detection of chromatin modifications within ~20-nm proximity of the histone posttranslational modifications in fixed cells. The authors combined several techniques, including proximity-induced DNA cleavage, rolling-circle amplification, fluorescence imaging, and micro-fluidic based cell capture, to achieve chromatin "nanoenvironment" profiling in a variety of cells, including those collected from cancer patients. Overall, this is an interesting and well thought-out study that will interest DNA nanotechnology, nucleus biology, and epigenetics community. The manuscript is well-written. I particularly liked the Discussion section where the authors articulate the limitations of the system. I have the following concerns that I want to see the authors address in a revised manuscript.

- (1) Experiment design: Overall, the study is well thought-out. The data quality is high and analyses are thorough. However, I find the following critical controls missing. On the DNA-origami substrate, an important negative control is a structure without walking probe (WP). Additionally, DNA origami is the perfect platform to establish distance-dependent Cell-TALKING and spatial resolution of the imaging method. Currently, these aspects are underdeveloped. For example, authors claimed longer probes may lead to longer-range recognition/detection, but did not systematically test that. Signal heterogeneity among different DNA-origami structures also warrants further study/explanation, because in cellular applications this may be interpreted as cell-to-cell variance. A related (and important) point is that the chromatin modification labeling efficiency needs to be quantified.
- (2) About the method/data presentation: Confocal microscopy is a good choice for proof-of-principle study, despite the limited resolution. However, authors presented the maximum brightness reconstruction images in most of the figures, which do not show the 3D information.

This makes it hard to access possible signal overlapping in 3D. The bioorthogonal labeling chemistry can be better explained, either in the main text or Supplementary Information. For example, the kinase activity of 5-HMUDK is not well explained. The sequential labeling of 5hmU, 5hmC, and 5fU can use a schematic diagram to help readers understand. The function of microfluidic device in this work needs to be clearly defined.

(3) Data interpretation: Figure S3B is a strong evidence for the specificity but a weak evidence for 5hmU labeling/detection efficiency. In figure 3A, two of the cell types appeared to have 5fU (green) spots enriched at/near the nuclear envelope, why? Overall, I find the cellular chromatin modification data hard to interpret. It is true that they generally make sense, but without direct comparison to the gold standard (e.g. CHIP-seq) or a complementary method (e.g. proximity ligation) it is hard to access the efficacy of DNA-TALKING technique. The manuscript can be strengthened by discussing the biological meanings of some results in more depth (e.g. in Figure 5A, the differential chromatin modifications in different cell lines.)

(4) Other: The authors correctly pointed out that the "one-to-many" indexing is a valuable feature of the DNA-TALKING technique. Unfortunately, this is not taken full advantage of in the current proof-of-principle applications. Most of the data suggest 1-to-1 proximity/interaction between a histone PTM and a DNA modification, and in the rare cases where one-to-many interactions are suspected in biological samples, the authors could not be certain because of the imaging resolution. I understand that super-resolution microscopy can technically demanding, but would appreciate a demonstration of feasibility, at least on the DNA-origami substrate (related to my first comment).

Reviewer #3 (Remarks to the Author):

We consider the technology developed in the paper to be sufficiently novel and useful for publication, however in our judgement the paper as-is requires a substantial number of further experiments and controls in order for the publication to be of sufficient quality and utility for the community. We also strongly recommend edits of figures and text for clarity, with the most important cases enumerated below.

Major/critical revisions:

Negative controls for Fig. 1A & 1B, i.e. DNA origami with only one target present, and the other target either missing or containing a scrambled annealing sequence.

Quantification for failure to overlap spots in Fig. 1A/1B and Fig. S1, as already pointed out by the authors in the text.

This is important for interpreting all downstream experiments (such as those in fixed cells), as it indicates something about the efficiencies of TALKING taking place for neighboring molecules. Otherwise, we don't know the false positive/false negative rates of TALKING signal even on designed DNA origami. Without this characterization, how are we to conclude that the absence of a particular signal at a location is due to genuine absence of some labeled DNA modification vs. the failure during TALKING? Fig. S1, for example, shows substantial variability in spot overlaps.

We recommend performing a sufficiently large number of experimental replicates with origami to characterize the frequency of how often TALKING fails to detect a particular target. This quantification should be kept in mind when interpreting all downstream experiments.

As above, quantify and characterize near-failure to overlap spots in Fig. 1C and Fig. S2. Although close examination of Fig. S2 visually confirms all spots are present, some—in fact, a substantial number as a proportion of all spots—of them are very faint. It is conceivable that repeating this experiment will result in one of the spots being undetectable. Again, this is critical to interpreting all downstream experiments.

Fig. 1D is unclear. Is it the case that the maximum distance that should be spannable is 7nm? In that case, how do we interpret the "leaky" positive signals detected at distances of 14 and 22 nm?

If, on the other hand, we are misinterpreting the claim, and it is indeed possible by design for the walking probe to reach 22 nm, then please rewrite for clarity.

In either case, then we recommend the authors should use experimental replicates of this DNA origami model to quantitatively characterize the tradeoff between distance and signal. Otherwise, in light of the admitted limitations of the microscopy resolutions used in the paper, it is hard to interpret data from downstream experiments.

Negative controls for the fixed cell experiments in Fig. 3.

What we mean here is to repeat the experiments, except with the blocked labels missing and/or with scrambled primer sites. This will confirm that the signals we observe are not spurious, and require all components to be present.

Another useful negative control we recommend is to first flush in labels missing the proper blocked DNA oligos and allow them to "label" or "occupy" all sites of interest. Then flow in proper labels for that site. None (or very few) should attach, and hence yield almost no signal. This would be a control against spurious labeling.

In multiple figures & analyses, such as in Fig. 3C, authors apply cross-correlations to one-dimensional cross-sections of puncta to quantify correlations. While this is valid in principle, it is less powerful and informative than using two-dimensional metrics for puncta overlap. Puncta colocalization analysis tools are regularly used in cell biology to look at e.g. colocalization of RFP- & GFP-labeled proteins. There are multiple metrics available. Tools such as ImageJ should have modules available.

Minor revisions:

Overall language and explanations need to be rewritten more clearly before publication.

Need a clearer diagram and explanation of the actual walking mechanism in Fig 1. This is a key concept & figure.

Fig. 1D: Please clarify what "Relative spot (%)" means, and how it's calculated. We have an idea of what the authors mean, however clarity here would help.

Make clear in Fig. 1 or main text description what the locked base actually is. Per supplement it is iXNA_G.

Explicitly point out if this walker design has been published before by authors themselves or others, or is it completely novel? If it is not completely novel, please provide a summary of differences and/or citations.

Fig. 3B and similar figures need to clarify what "Spot Percentage (%)" means and how it's quantified. The description as-is is not completely transparent.

Point-by-point response to the reviewers' comments for NCOMMS-20-27973

(Title: Cellular Macromolecules-Tethered DNA Walking Indexing to Explore Nanoenvironments of Chromatin Modifications)

Reviewer: 1

In this manuscript, the authors reported a newly developed method (Cell-TALKING) to explore nanoenvironments of chromatin modifications, which was applied in breast cancer cell lines and clinical specimens. This methodology can be used for analyzing single-cell chromatin nanoenvironments, where the modification sites in the nanoenvironment are labeled with respective DNA barcodes by Cell-TALKING, and then visualized by rolling circle amplification (RCA)-improved fluorescence imaging. The method itself is intriguing, however, there are several major issues:

(1) The method relies on many sequential steps and it is currently unclear what is the level of non-specific signal that accumulates? There are several controls that are missing in order to understand the expected signal/noise ratio?

Response: Thanks for the valuable comment! In Figure S4, we have demonstrated negligible non-specific background signals induced by several negative controls which were in the absence of enzymes (5-HMUDK or T4 β -GT), antibodies or walking probes. As the reviewer suggested, we also supplemented other negative controls to clearly investigate the accumulated non-specific signal. They include the one without the nicking enzyme, the one with the mismatched RCA probes and the one that pre-labels modification sites with blocked duplex barcoding probes. And the distribution of single-cell spot counts of both these negative controls and a positive control were also depicted to indicate the signal-to-noise level of our method. As shown in revised Figure S4, only the positive control presented high values of single-cell spot counts, whereas all negative samples induced nearly zero or very low spot counts. These results indicated the negligible non-specific signal accumulated by our Cell-TALKING method. A relevant description was also added in the main text of the revised manuscript.

Figure S4. Specific cell imaging using Cell-TALKING. (A) Scheme of sequential labeling of 5hmU, 5hmC and 5fU followed by the histone PTM in cells. The blocked walking probe is a DNA hybrid duplex that can be captured by the DNA-crosslinked secondary antibody. The detail is described in the protocol of Cell-TALKING as below. (B) Selected cell images for several negative controls and one positive sample. The nanoenvironment around histone H3K27ac in MCF-10A cells was detected here. Scale bar, 10 μm . (C) The spot counts of three combination sites of single cells (cell numbers are 50). The “Non-walking probe” negative control used the Non-walking probe instead of the walking probe. The Non-walking probe can not hybridize to all three barcoding probes, which fails to induce the DNA nicking reaction and walking process. The “Mismatched probes” negative control used a circularized padlock that can not hybridize to all three barcoding probes, which fails to induce RCA. And the “Pre-block modification sites” control firstly labeled all modification sites with blocked duplex barcoding probes, and then with single-stranded barcoding probes. The blocked duplex barcoding probes were prepared by the hybridization of the complementary oligo to the single-stranded barcoding probes. They can not hybridize to their circularized padlocks and fail to induce RCA. This negative control can indicate that almost all sites were labeled and crosslinked with DNA probes by the first labeling processes, and no residual sites were detected by second labeling processes. Similar results are shown in Figure S3.

(2) The method relies on antibodies, similarly to ChIP-seq. Epitope occlusion is a major issue in antibody recognition in these techniques and I have not seen controls to address the issue or any discussion of it in the manuscript.

Response: Thanks very much for pointing out this issue! As recommended, we have discussed the epitope occlusion about PTM antibody recognition used in our method. A relevant description was added in the Discussion section of the revised manuscript as below:

“Furthermore, PTMs recognition via antibodies used in our method may be affected by epitope occlusion.⁴⁶ The specific binding of antibodies to a single histone PTM could be obfuscated by any neighboring PTMs at the same tail. This steric occlusion effect is a common issue for PTMs recognition, especially on highly modified histones. It requires to generate more specific antibodies against histones containing various modification valences.^{47”}

(3) As the authors mention, they have recently published several similar techniques probing epigenetic modifications in single cell. It seems that the major advancement in this paper is the capacity for multiplexing, but it is not clearly indicated. If the leap is in multiplexing, the authors should provide new epigenetic insights or novel biology to justify publication in Nature Communications.

Response: Thanks for the comment! As we described in the Introduction section of the manuscript, we and others have reported fluorescence detection methods to probe single or two epigenetic modifications in single cells. By designing parallel and orthogonal probe sets, these previous methods could be routinely expanded for multiplexed detection of three or more independent modifications that are not spatially colocalized.

However, they and their potential multiplexed versions are unable to probe the

spatial compositions and relationships of multiple modifications within nanoscale regions termed nanoenvironments, such as the regions of < 20 nm radius around one histone or one PTM. The nanoenvironment of chromatin modifications is a new concept and is different from alone, discrete modifications. Therefore, we developed the Cell-TALKING method based on dynamic DNA nanotechnology and barcoding amplification in this paper. As far as we know, for the first time, our method achieved the probing of nanoenvironments of ~ 20 nm radius in cells, in despite of some remaining challenges (discussed in the Discussion section of the manuscript). Therefore, the leap of our manuscript is developing novel Cell-TALKING method to probe cellular nanoenvironments rather than revising previous methods for parallel multiplexing. To clearly indicate this major advancement, we revised the Introduction section together with Discussion section, and emphasized this point in the revised manuscript as below:

“These methods could be routinely expanded for parallel detection of three or more spatially discrete, alone modifications by designing orthogonal probe sets, yet they are unable to probe spatial relationships of multiple modifications within nanoenvironments.”

“As described in the introduction, the analysis of cellular nanoenvironments is quite different to both the detection of two proximity targets by PLA and even the parallel PLA analysis. Very recently, we reported the spatial proximity of 5fC and 5hmC based on an improved PLA.⁴⁵ However, PLA can't probe the nanoenvironments of chromatin modifications as discussed above. Especially, our Cell-TALKING method is unable to detect discrete, alone modification sites that are not in the nanoenvironments. Thus new methods are required to differentially detect the sites within or outside the nanoenvironments in the same single cells.”

(4) The physiopathological relevance of the chromatin nanoenvironment reported by the authors and the role it plays in cancer development remain unclear and should be better discussed.

Response: Thanks for the valuable comments! The major advancement in our paper is

to develop novel Cell-TALKING method to probe cellular nanoenvironments for the first time. To demonstrate this method, we chose chromatin modifications as detection biomolecule targets and used cancer cell lines together with clinical specimens as testing samples. The detection data have been depicted in Figure 5. As suggested by the reviewer, we made a further discussion about these results in the revised manuscript as below:

“Next, we characterized the nanoenvironments of chromatin modifications in different cancer cell lines. Three human breast cell lines, including nontumorigenic MCF-10A, carcinogenic MCF-7 and highly invasive MDA-MB-231 were used as examples. We detected three DNA modifications around H3K4me3, and the merged images of cells were shown in Figure 5A. These cell lines had different levels of three modification combinations according to their single-cell spot count and fluorescence intensity (Figure S9). The averaged value of 5hmU/H3K4me3 combination in MDA-MB-231 is lower than that in MCF-10A or MCF-7, possibly implying the potential negative effect of 5hmU/H3K4me3 combination on cancer metastasis. Other two combinations do not present remarkable changes in these three cell lines. Furthermore, the distances of each spot to the nucleus periphery were also calculated by MATLAB (Figure 5B). We observed that the distances of all three combinations in MCF-10A are lower than those in MCF-7 and MDA-MB-231. It suggests such distances of these modification combinations might be related to cancer progression and metastasis.

Moreover, we applied Cell-TALKING to clinical specimens with the integration of a microfluidic chip (Figure 5C). Microfluidic techniques have been widely used to capture, culture or/and fix rare cells. We designed a simple chip as an example for following experiments mainly including cell culture, fixation, reaction and imaging. Cell suspensions derived from tissues or FNA biopsies of lung cancer patients were tested. We still detected the nanoenvironments containing 5hmU, 5fU and 5hmC around H3K4me3. As shown in Figure 5C, averaged spot counts of 5hmU/H3K4me3 combination were higher than those of 5hmC/H3K4me3 or 5fU/H3K4me3 in these cancer samples. It may imply potential positive function of 5hmU/H3K4me3 combination in cancer. In contrast, quite low levels of 5hmC/H3K4me3 combination in most cancer samples may suggest its potential negative correlation with cancer. Despite these quite simple exploration, systematic experiments are required to validate those superficial speculations and to further clarify

the relevance of chromatin modification nanoenvironments to cancer or other diseases.^{20, 41} In

principle, Cell-TALKING may also be used to analyze tissue sections with routine manipulation processes. Detecting section samples may identify cell subtypes and uncover the spatial relationship between different single cells. In addition, our method can be applied to other samples such as embryo stem cells or nerve cells to explore the function of combinational chromatin modifications in differentiation and development.”

(5) While the method provides information of proximity, there is still no validation that marks that are detected to be close in space by this method are in fact, for example, on the same tail and represent an epigenetic state.

Response: Thanks for the valuable comment! Spatial colocalization or proximity of diverse biomolecules or components in cellular environments underpin the fundamental processes of life. During the past decades, accumulated evidence has confirmed extensively existing spatial proximity or interaction of protein/protein, RNA/protein, DNA/protein, DNA/RNA or DNA/DNA. Especially, chromosome conformation capture technologies such as Hi-C sequencing and its derivatives have been developed to investigate DNA/DNA interaction or 3D genomic conformation (Nature Genetics, 2015, 47, 598-609; Nature Genetics, 2018, 50, 754-76347; Cell, 2018,174, 1-14; Nature Methods, 2018, 15, 741-747; Nature, 2020, 586, 139-144; Nature Biotechnology, 2020, doi.org/10.1038/s41587-020-0643-8.). Numerous studies found complex chromatin interactions or spatial proximity in cell nucleus. According to this finding, we speculate that diverse chromatin modifications (DNA modifications and histone PTMs), which are located at the same genomic region or chromatin and even between different genomic regions or different chromatins, may co-exist in the same cellular nanoenvironment such as the regions of < 20 nm radius around one histone. However, now there are no available detection methods to probe such a cellular nanoenvironment.

To address this issue, we developed a proof of concept method Cell-TALKING to probe such nanoenvironments on DNA origamis and in cells for the first time. To

demonstrate this method, we imaged diverse DNA modifications (5hmU, 5hmC and 5fU) within the regions of < 20 nm radius around one histone PTM. And the detected proximity sites within the nanoenvironments by our method may exist on different genomic regions and even chromatins or on the same one. We didn't investigate two or more histone PTMs in the same nanoenvironments in this work. As discussed in the Discussion section, our method in principle can be used to detect diverse TFs or histone PTMs in the same nanoenvironments. Notably, we can't validate whether the detected spatial proximity PTMs (or sites) locate on the same tail (or the same genomic region for DNA marks) or not. And as far as we know, there is no existing methods to determine this issue. We will appreciate the reviewer very much for telling us feasible solutions, which could help improve and clarify our manuscript greatly.

In addition, to investigate whether the detected spatial proximity marks in cellular nanoenvironments represent new epigenetic states and even biological functions, it requires professional and systematic biological experiments for deeper studies. That will be another significant but difficult work. Overall, to clearly explain this issue, we has added a related description in the Discussion section of the revised manuscript as below:

“Nevertheless, the detected combinatorial chromatin modifications within the nanoenvironments may locate on different genomic regions and even different chromatins ascribed to complex chromatin interaction. And their epigenetic functions remain undefined.”

(6) For a methodology paper, the exact methodological process and protocol are not clear or detailed enough and an elaborated materials and methods should be included in the supplementary information.

Response: Thanks for the positive comment and suggestion! As recommended, we have added a detailed, step-by-step protocol of our method Cell-TALKING in the revised Supplementary information.

Reviewer: 2

The manuscript reports a new method for the multiplexed detection of chromatin modifications within ~20-nm proximity of the histone posttranslational modifications in fixed cells. The authors combined several techniques, including proximity-induced DNA cleavage, rolling-circle amplification, fluorescence imaging, and micro-fluidic based cell capture, to achieve chromatin "nanoenvironment" profiling in a variety of cells, including those collected from cancer patients. Overall, this is an interesting and well thought-out study that will interest DNA nanotechnology, nucleus biology, and epigenetics community. The manuscript is well-written. I particularly liked the Discussion section where the authors articulate the limitations of the system. I have the following concerns that I want to see the authors address in a revised manuscript.

(1) Experiment design: Overall, the study is well thought-out. The data quality is high and analyses are thorough. However, I find the following critical controls missing. On the DNA-origami substrate, an important negative control is a structure without walking probe (WP).

Additionally, DNA origami is the perfect platform to establish distance-dependent Cell-TALKING and spatial resolution of the imaging method. Currently, these aspects are underdeveloped. For example, authors claimed longer probes may lead to longer-range recognition/detection, but did not systematically test that.

Signal heterogeneity among different DNA-origami structures also warrants further study/explanation, because in cellular applications this may be interpreted as cell-to-cell variance.

Response: Thanks for the valuable suggestions! We have supplemented related experiments on DNA origami substrates as suggested by the reviewer.

For the first point, the results were shown in revised Figure S2. The negative controls without probes (walking probe or target barcoding probe) or with mismatched barcoding probes induced very low fluorescence response (bottom of Figure S2A), indicating negligible false positive signals on designed DNA origami substrates.

For the second point, we used other walking probes containing a longer spacer (additional 15 nt or 30 nt) to probe the long-distance detection site (22 nm). We found that long walking probes increased the signals at 22 nm detection site (Figure S2B). It demonstrated the probe length-determined recognition/detection range of our method. Furthermore, we also supplemented more negative control experiments for cell imaging analysis as suggested by Reviewer 1 and 3, results of which were presented in revised Figure S4. The distribution level of single-cell spot counts of both negative controls and a positive sample was also depicted in that figure to indicate the signal-to-noise level of our method.

For the third comment, we quantified the number of spots with fluorescence overlap or not by five experimental replicates (Figure S1 and S2A). About 50 images were count in each experimental replicate. Substantial heterogeneity in spot fluorescence overlap was observed. Some spots only present one or two fluorescence channels rather than all three, though the DNA origami substrate used here contains all three target barcoding probes. Such a fluorescence overlap heterogeneity indicated the false negative signals on DNA origami substrates. This false negative result may be mainly ascribed to both the limited efficiencies of reactions performed on DNA origami-modified coverglass and the failure of the formation of DNA origami substrate. It is well known that molecular reactions such as DNA hybridization and enzymatic reactions are of low efficiency at water/glass interface compared to in solutions, and DNA hybridization on DNA origami is restricted by strong electrostatic repulsion. Also, the DNA origami structures rarely have a 100% formation yield (Chemical Reviews, 2017, 117, 12584-12640). Their yield depends on the complexity of design and experimental conditions. Various incorrectly folded structures often appear among the final products (Nature Methods, 2020, 17, 789-791; Angew. Chem. Int. Ed. 2020, 59, 2-9.).

In the paper, our purpose is to detect cell samples, and we used DNA origami substrates to demonstrate our method. Still, all these supplementary experiments well confirm the performance of our proposed method and help improve our paper. Now a related explanation about these results was added in the revised manuscript as below:

“We also designed different two-dimensional DNA origami structures to verify TALKING. It is well known that nanoscale distances between different sites on DNA origami are easily controlled and designed. Firstly, we designed three types of barcoding probes surrounding the walking probe with about 7 nm radius distance as an example. These DNA probes were attached on the origami by DNA hybridization, and the origami was fixed on a coverslip (Figure 1C). Atomic force microscopy was used to confirm our DNA origami structure (Figure 1C, bottom left). The fluorescence signals of three channels overlapped (Figure 1C and S2A), which was consistent with our results in Figure 1B. In contrast, negative controls without probes or with mismatched probes presented very low fluorescence response (Figure S2A), indicating negligible false positive signals on the origami substrates. Notably, substantial heterogeneity in spot fluorescence overlap was observed. Thus, we quantified the number of spots with fluorescence overlap by five experimental replicates (Figure S2A). A portion of spots missed one or two of all three fluorescence signals, indicating the false negative level of our method on this origami substrate. This false negative result may be ascribed to both the limited efficiencies of reactions on coverglass and the failure of the formation of DNA origami substrates. It is well known that molecular reactions such as DNA hybridization and enzymatic reactions are of low efficiency at water/glass interface compared to in solutions. DNA hybridization on DNA origami is also restricted by strong electrostatic repulsion. The yield of DNA origami structures rarely reaches 100%, mainly depending on the complexity of design and experimental conditions. The failure of fluorescence overlap of spots in Figure 1A and 1B can be explained similarly. Notably, some spots presented relatively large sizes, which may result from non-uniform RCA efficiency at this interface or the aggregation of multiple copies of DNA substrate molecules.

We further investigated the distance effect on proximity recognition of our method. DNA origami structures were designed to attach one barcoding probe with different separation distances (about 7 nm, 14 nm, or 22 nm) to the walking probe or target site (Figure 1D). The number of spots were reduced as the increase of separation distances. The normalized spot counts of the samples of other distances to that of 7 nm were shown in the bottom right panel of Figure 1D. In principle, the maximum recognition distance could reach the sum of the length of barcoding probe and that of walking probe. However, the stretching of flexible DNA may be restricted at water/glass interface or by strong electrostatic repulsion from DNA origami. Increasing the length of probes may extend the recognition distance. Here we used longer walking probes containing an additional spacer (15 nt or 30 nt) to investigate this issue. As shown in Figure S2B, longer walking probes increased the signal response at long-distance detection sites. This result confirmed the probe length-determined recognition range of our method. Therefore, our method achieved one-to-many proximity recognition and indexing performance. These results may enable the exploration of combinatorial patterns of diverse chromatin modifications in the same nanoenvironments.”

Figure S2. Performance of *in vitro* Cell-TALKING on DNA origami substrates. (A) Upper, the actual walking mechanism. Two copies of three barcoding probes (BPs) surround the blocked walking probe (WP) with about 7 nm radius distance on one origami molecule. During the nicking/walking process, the nicking enzyme firstly cuts the blocker sequence in the blocked WP duplex. Then the walking probe molecule in this duplex can be isothermally released to hybridize one BP molecule, forming a DNA duplex as the substrate for nicking enzyme. After cutting this BP molecule in the duplex, the walking probe molecule will walk to and hybridize another BP molecule. In this way, all these BP molecules can be cleaved and the newly generated 3'-OH ends can induce RCA reaction. Middle, analysis of false negative response of our method. The images

are corresponding to Figure 1C. Bottom, investigating the false positive response by several negative controls. There are some spots that fail to overlap two or three fluorescence signals, indicating the false negative response of our method on this DNA origami substrate. It is discussed in the main text. (B) Investigating the probe length-determined recognition range of our method using different walking probes and DNA origami substrates. Five repeated experiments were performed for statistical analysis of spot count of randomly selected images. The “normalized spot count” means the spot count of one sample relative to that of the given one (e.g., the positive control (IV) or the sample using walking probe containing 0 nt spacer).

A related (and important) point is that the chromatin modification labeling efficiency needs to be quantified.

Response: Thanks for the valuable comment! As the reviewer described, a quantification analysis of the labeling efficiency of chromatin modifications can better demonstrate the performance of our proposed method and greatly improve our paper. In our work, the labeling process of DNA modifications in fixed cells included two steps: first labeling with small molecules and second crosslinking with DNA primers (barcoding probes). Then the crosslinked DNA primers within the same nanoenvironments will be cleaved, and then induce barcoding RCA for following DNA hybridization fluorescence imaging. It generated fluorescent spots for the imaging analysis. We obtained the number of fluorescent spots, total signal intensity of single cells and subcellular distribution information. But the number of fluorescent spots is not able to exactly represent the total abundance or number of modification sites as we discussed in the main text about Figure 3B.

To quantitatively assess the labeling yield of chromatin modifications with the DNA primers, we should detect both the number of total modification sites and the number of DNA primer-labeled sites. It is well known that the total number of modification sites of bulk genomic DNA materials can be detected by ultrasensitive analytical liquid chromatography-tandem mass spectrometry coupled with chemical

isotopic labelling (Nature Chemical Biology, 2014, 10, 574-581; Nature Chemistry, 2014, 6, 1049-1055; Chemical Communications, 2016, 52, 737-740.). This mass spectrometry method required roughly more than 10 µg input of genomic DNA (10^6 - 10^7 cells) especially for low-abundance chromatin modifications. But it is still difficult to quantify the DNA primer-labeled sites by mass spectrometry method mainly due to the lack of isotopic reagents for these labeled products.

On the other hand, the modification sites were crosslinked with DNA primer with a 1:1 stoichiometry in our work. Thus the molecule or copy number of DNA primer can represent the number of DNA primer-labeled sites. To quantitate the molecule number of a specific DNA sequence, the quantitative real-time PCR (qPCR) is a well-known method based on a standard curve of a serial of known concentrations of target DNA sequence. The DNA sequence can be exponentially amplified by qPCR as the template, and its length should be over 100 nt for efficient DNA amplification.

In the past decade, the next-generation sequencing (NGS) technique has been widely used to detect genomic and transcriptomic sequences. And the unique molecular identifiers (UMIs), random sequences that label individual molecule, have been introduced to count the copy numbers of genes or RNA molecules before PCR amplification (Nature Methods, 2014, 11, 163-166; Cell, 2015, 161, 1202-1214.). We suggest UMI-assisted NGS may detect the copy number of DNA primer labeled at modification sites with high throughput (e.g. detect multiple chromatin modifications simultaneously).

In summary, the quantitative assessment of the degree of labeling of modification sites with the DNA primer may be realized by the combination of above methods that are based on different mechanisms. Notably, due to the highly different detection performance between mass spectrometry method and qPCR or NGS, such a quantitative assessment result may be not very accurate. We also appreciate available and workable solutions from the reviewer, as addressing this point can help improve and clarify our manuscript greatly.

(2) About the method/data presentation: Confocal microscopy is a good choice for proof-of-principle study, despite the limited resolution. However, authors presented the maximum brightness reconstruction images in most of the figures, which do not show the 3D information. This makes it hard to access possible signal overlapping in 3D.

The bioorthogonal labeling chemistry can be better explained, either in the main text or Supplementary Information. For example, the kinase activity of 5-HMUDK is not well explained.

The sequential labeling of 5hmU, 5hmC, and 5fU can use a schematic diagram to help readers understand.

The function of microfluidic device in this work needs to be clearly defined.

Response: Thanks for the valuable comments!

For the first point, as the reviewer described, we used confocal microscopy for proof-of-principle study, despite its limited resolution. As described in the section of “Imaging and data analysis”, we imaged the cells in 3D with the z-stack at intervals of 0.5 μm , and the resulting z-stack images were combined to a single one by maximum intensity projection using LAS X Version of the microscope. These maximally projected images of cells can be used for the extraction of single-cell fluorescence intensity and spot count by using custom software written in Matlab, ImageJ plug-in or any other image-processing algorithms. We do this processing following the instructions of the softwares, and it is consistent with those of previous studies (Nature Protocols, 2012, 7, 408-419; Nature Protocols, 2013, 8, 1743-1758; Chem, 2018, 4, 1-14.).

We also roughly depicted the fluorescence overlap of spots according to a large distance of about 400 nm in Figure 3C, in which only the maximally projected images were shown. As suggested by the reviewer, we supplemented the Z-stack 3D image analysis of the image used in Figure 3C as an example to present possible signal overlap in Z-stack 3D. The result was shown in revised Figure 3C and Figure S7 in the revised Supplementary information. We can see that the spots highlighted in circle are overlapped in two or three fluorescence channels with the resolution of hundreds

of nanometers. Notably, some overlapped spots in maximally projected images are not overlapped in 3D, and the data in Figure 3C are conceptually and roughly presented based on above-mentioned conditions, which has also been described in the main text of the manuscript. Thus we didn't summarize numbers of the spots that were overlapped in 3D or 2D again. Especially, strict and systematic experiments are required to verify above results.

Super-resolution microscopy is a better choice than confocal microscopy to detect the colocalization of two or three targets. But it often suffers from the registration errors or positioning errors from sample drift in multichannel imaging and low imaging throughput, and its performance can vary considerably in optically complex samples. In principle, other DNA proximity recognition techniques similar to that used in our method can be developed to directly detect the nanoscale colocalization of more than three given targets. For example, we can design a complex DNA structure or substrate that will only be formed in the presence of all the DNA recognition probes within the same nanoenvironment. And this DNA formation product can only trigger one signal that indicates the colocalization of those targets. Such a design and study may be another interesting work.

Figure 3. Exploring different nanoenvironments of chromatin modifications in single cells. (A) Selected merged cell images for five histone PTMs. The red, green and blue fluorescence spots each indicated the combination sites of 5hmU/PTM, 5fU/PTM and 5hmC/PTM. The scale bar is 5 μm . (B) The spot percentages of three combination sites of single cells in five samples (cell numbers = 52, 34, 32, 43 and 38, respectively.). (C) Analysis of the overlap or cluster of different spots.

Figure S7. 3D overlap analysis of the spots from different fluorescence channels in the image used in Figure 3C. The images were acquired in 3D with the z-stack at intervals of 0.5 μm . The spots highlighted in circle are overlapped in three or two fluorescence channels with the spatial resolution of hundreds of nanometers.

For the second point, we have explained the kinase activity of 5-HMUDK for the bioorthogonal labeling of 5hmU in DNA in the revised manuscript as below:

“We have previously reported the differentiated labeling of 5hmU and 5hmC.²⁰ The 5hmU sites in dsDNA was firstly labeled with a phosphate or thiophosphate group by 5-hydroxymethyluridine DNA kinase (5-HMUDK), and then the 5hmC was labeled with an azido group by T4 Phage β -glucosyltransferase (T4 β -GT). 5-HMUDK can catalyze the transferring of the γ -phosphate of adenosine triphosphate (ATP) to the hydroxymethyl moiety of 5hmU in dsDNA. Nevertheless, abundant natural phosphate groups or thiol groups exist in cells, and the bioorthogonal labeling of 5hmU by 5-HMUDK remains a challenge. An ATP analogue containing bioorthogonal crosslinking groups is required.”

For the third point, we have depicted the sequential labeling of 5hmU, 5hmC, and

5fU in cells, which is shown in Figure S4A in the revised Supplementary information.

For the fourth point, the design and function of microfluidic device used in this work have been clearly described in the experimental section as below:

“Design and fabrication of microfluidic chip. The microfluidic chip was designed by a computer-assisted design (CAD) software and the cast mould (50- μ m-thick) was prepared by photolithography using AZ 50XT positive photoresist on a polished silicon wafer. The chip was controlled by torque-activated valves, consisting of two layers of poly(dimethylsiloxane) (PDMS, oligomer and cross-linker at a ratio of 10:1 (w/w)). In brief, a 100- μ m-thick PDMS layer was spin-coated onto the cast mould to form the channels of the chip. After curing at 70 °C for 30 min, the hex nuts and nylon screws were embedded in another uncured PDMS layer over corresponding positions of the channels, then followed by the final curing step. The separate inlets and outlets, combining with torque-activated valves, function as two independent in and out systems for clinical samples and reagents. The chamber supports cell culture, subsequent fixation and Cell-TALKING reactions, also provides the flexible place for cofocal imaging.”

(3) Data interpretation: Figure S3B is a strong evidence for the specificity but a weak evidence for 5hmU labeling/detection efficiency.

In figure 3A, two of the cell types appeared to have 5fU (green) spots enriched at/near the nuclear envelope, why?

Overall, I find the cellular chromatin modification data hard to interpret. It is true that they generally make sense, but without direct comparison to the gold standard (e.g. CHIP-seq) or a complementary method (e.g. proximity ligation) it is hard to access the efficacy of DNA-TALKING technique.

The manuscript can be strengthened by discussing the biological meanings of some results in more depth (e.g. in Figure 5A, the differential chromatin modifications in different cell lines.)

Response: Thanks for the valuable comments!

For the comment about 5hmU labeling/detection efficiency, a similar response has been described above for answering the fourth point of Comment No.(1). And as we discussed in the main text about Figure 3B, the exposed or accessible modification sites were easily labeled and detected. Some spots presented relatively large sizes. They may suggest the adjacency of multiple modification sites or result from non-uniform RCA efficiency in the molecularly crowded cell nucleus. Therefore, a fluorescence spot may consist of one or more modification sites. The spot numbers or signal intensities of single cells may reflect not only the levels of modification sites but also their accessibility to the molecular probes or enzymes. This is a common issue in cell imaging analysis.

For the comment about 5fU (green) spots enriched at/near the nuclear envelope in Figure 3A, it may be ascribed to the cell-to-cell heterogeneity. As we know, single cells in cultured populations are quite different in the shape, phenotype, genotype, cell state and/or fate. Notably, we imaged lots of single cells from cultured cell populations and randomly selected many images for statistical analysis. And an image was used in Figure 3A. Other images of those two sample types (H3K27ac and γ H2AX) are shown below:

Other images of single cell for H3K27ac sample and γ H2AX sample related to Figure 3A.

In most of these images, the fluorescence signals are distributed throughout the

nuclear volume, and the signal heterogeneity is obvious. In addition, it was reported that partial chromatin regions such as late-replicating chromatin is in proximity to the nuclear lamina and nuclear envelope (Nature Reviews Molecular Cell Biology, 2019, 20, 721-737.). Thus, the fluorescence signals of chromatin modifications may appear near the nuclear envelope in certain cells due to heterogeneity.

For the comment about hard to interpret cellular chromatin modification data, we have supplemented proximity ligation assay (PLA) of the cell samples in comparison with our method. Here we selected two combination modifications, 5hmC/ H3K27ac and 5hmC/H3K4me3 as examples. The results were shown in Figure S5 of revised Supplementary information. We observed that similar detection results were provided by these two methods in both testing targets. It confirmed that the efficacy of our method was well consistent with that of PLA. However, PLA can't probe the nanoenvironments of chromatin modifications as explained in the Discussion section. A related description has been added in the main text of the revised manuscript.

Figure S5. Comparison of PLA and Cell-TALKING in the detection of single combination modification. (A) Selected merged cell images. The scale bar is 10 μm . (B) The spot counts of single cells from these samples (cell numbers are about 50). It can be observed that similar detection results were provided by these two methods in both testing targets. It confirmed that the efficacy of our method was well consistent with that of PLA.

For the comment about discussing the biological meanings of some results in Figure 5A, we made a further discussion about these results in the revised manuscript

as suggested by the reviewer. The detail is shown below:

“Next, we characterized the nanoenvironments of chromatin modifications in different cancer cell lines. Three human breast cell lines, including nontumorigenic MCF-10A, carcinogenic MCF-7 and highly invasive MDA-MB-231 were used as examples. We detected three DNA modifications around H3K4me₃, and the merged images of cells were shown in Figure 5A. These cell lines had different levels of three modification combinations according to their single-cell spot count and fluorescence intensity (Figure S9). The averaged value of 5hmU/H3K4me₃ combination in MDA-MB-231 is lower than that in MCF-10A or MCF-7, possibly implying the potential negative effect of 5hmU/H3K4me₃ combination on cancer metastasis. Other two combinations do not present remarkable changes in these three cell lines. Furthermore, the distances of each spot to the nucleus periphery were also calculated by MATLAB (Figure 5B). We observed that the distances of all three combinations in MCF-10A are lower than those in MCF-7 and MDA-MB-231. It suggests such distances of these modification combinations might be related to cancer progression and metastasis.

Moreover, we applied Cell-TALKING to clinical specimens with the integration of a microfluidic chip (Figure 5C). Microfluidic techniques have been widely used to capture, culture or/and fix rare cells. We designed a simple chip as an example for following experiments mainly including cell culture, fixation, reaction and imaging. Cell suspensions derived from tissues or FNA biopsies of lung cancer patients were tested. We still detected the nanoenvironments containing 5hmU, 5fU and 5hmC around H3K4me₃. As shown in Figure 5C, averaged spot counts of 5hmU/H3K4me₃ combination were higher than those of 5hmC/H3K4me₃ or 5fU/H3K4me₃ in these cancer samples. It may imply potential positive function of 5hmU/H3K4me₃ combination in cancer. In contrast, quite low levels of 5hmC/H3K4me₃ combination in most cancer samples may suggest its potential negative correlation with cancer. Despite these quite simple exploration, systematic experiments are required to validate those superficial speculations and to further clarify the relevance of chromatin modification nanoenvironments to cancer or other diseases.^{20, 41} In principle, Cell-TALKING may also be used to analyze tissue sections with routine manipulation processes. Detecting section samples may identify cell subtypes and uncover the spatial relationship between different single cells. In addition, our method can be applied to other samples such as embryo stem cells or nerve cells to explore the function of combinational chromatin

modifications in differentiation and development.”

(4) Other: The authors correctly pointed out that the "one-to-many" indexing is a valuable feature of the DNA-TALKING technique. Unfortunately, this is not taken full advantage of in the current proof-of-principle applications. Most of the data suggest 1-to-1 proximity/interaction between a histone PTM and a DNA modification, and in the rare cases where one-to-many interactions are suspected in biological samples, the authors could not be certain because of the imaging resolution.

I understand that super-resolution microscopy can be technically demanding, but would appreciate a demonstration of feasibility, at least on the DNA-origami substrate (related to my first comment).

Response: Thanks for the valuable comment! As the reviewer discussed and recommended, we try our best to image multiple fluorescence responses by one walking probe molecule on the DNA origami substrate induced by our proposed method via super-resolution microscopy. Here we used an inverted Leica TCS SP8 STED 3X microscope (Leica Microsystems, Mannheim Germany) equipped with a 100x/1.4 NA oil immersion objective. As multiplexed fluorescence imaging by this super-resolution microscope is quite difficult to achieve, we select two fluorescence channels (FAM and Cy3) to investigate one-to-many recognition by our method.

In this design, we used a simple DNA origami substrate containing two detection sites with a large distance of about 40 nm as an example. The size of the DNA origami is about 90 nm. After full reactions of our method, these two detection sites induced green (FAM) and yellow (Cy3) signals, respectively. Fluorescence channels (FAM or Cy3) used excitation wavelength 488 nm or 545 nm (laser light derived from an 80 MHz pulsed White Light Laser), and the stimulated emission was performed with a 592 nm pulsed laser. For FAM and Cy3 dual color imaging, two spectral windows of typically 495-540 nm and 565-620 nm were applied. Strong photobleaching was observed during STED imaging.

The image is shown below. The sizes of fluorescence spots are roughly about 80-100 nm, and two spots are partially overlapped. According to the size of one DNA origami molecule (about 90 nm) and the distance between two detection sites (about 40 nm), we can conclude that the overlapped two fluorescence signals are on the same DNA origami molecule, which demonstrates the one-to-two recognition performance of our method on DNA origami substrates. In principle, using more advanced super-resolution microscopy with higher resolution can visualize two completely separated spots on the same copy of this DNA origami substrate.

Fluorescence image of the DNA origami substrate containing two detection sites (~ 40 nm distance) by STED super-resolution microscopy.

Reviewer: 3

We consider the technology developed in the paper to be sufficiently novel and useful for publication, however in our judgement the paper as-is requires a substantial number of further experiments and controls in order for the publication to be of sufficient quality and utility for the community. We also strongly recommend edits of figures and text for clarity, with the most important cases enumerated below.

Major/critical revisions:

Negative controls for Fig. 1A & 1B, i.e. DNA origami with only one target present, and the other target either missing or containing a scrambled annealing sequence.

Quantification for failure to overlap spots in Fig. 1A/1B and Fig. S1, as already pointed out by the authors in the text.

This is important for interpreting all downstream experiments (such as those in fixed cells), as it indicates something about the efficiencies of TALKING taking place for neighboring molecules. Otherwise, we don't know the false positive/false negative rates of TALKING signal even on designed DNA origami. Without this characterization, how are we to conclude that the absence of a particular signal at a location is due to genuine absence of some labeled DNA modification vs. the failure during TALKING? Fig. S1, for example, shows substantial variability in spot overlaps.

We recommend performing a sufficiently large number of experimental replicates with origami to characterize the frequency of how often TALKING fails to detect a particular target. This quantification should be kept in mind when interpreting all downstream experiments.

As above, quantify and characterize near-failure to overlap spots in Fig. 1C and Fig. S2. Although close examination of Fig. S2 visually confirms all spots are present, some-in fact, a substantial number as a proportion of all spots-of them are very faint. It is conceivable that repeating this experiment will result in one of the spots being undetectable. Again, this is critical to interpreting all downstream experiments.

Response: Thanks for the valuable comments. As recommended, we have

supplemented related experiments. The results were shown in Figure S1 and S2, and were discussed in the main text in revised manuscript. All these revisions help improve our paper.

For adding negative controls, the results were shown in Figure S2A (bottom). The negative controls without probes (walking probe or target barcoding probe) or with mismatched barcoding probes induced very low fluorescence response, indicating negligible false positive signals on designed DNA origami substrates. We also supplemented more negative control experiments for cell imaging analysis. The related results were presented in Figure S4B and S4C. The distribution level of single-cell spot counts of both negative controls and a positive sample was also depicted. It indicated very low false positive response for cell imaging by our method.

For the comment about characterizing failure to overlap spots, we quantified the number of spots with fluorescence overlap or not by five experimental replicates (Figure S1 and S2A). About 50 images were count in each experimental replicate. Substantial heterogeneity in spot fluorescence overlap was observed. Some spots only present one or two fluorescence channels rather than all three (Figure S2A), though the DNA origami substrate used here contains all three target barcoding probes. Such a fluorescence overlap heterogeneity indicated the false negative signals on DNA origami substrates. This false negative result may be mainly ascribed to both the limited efficiencies of reactions performed on DNA origami-modified coverglass and the failure of the formation of DNA origami substrate. It is well known that molecular reactions such as DNA hybridization and enzymatic reactions are of low efficiency at water/glass interface compared to in solutions, and DNA hybridization on DNA origami is restricted by strong electrostatic repulsion. Also, the DNA origami structures rarely have a 100% formation yield (Chemical Reviews, 2017, 117, 12584-12640). Their yield depends on the complexity of design and experimental conditions. Various incorrectly folded structures often appear among the final products (Nature Methods, 2020, 17, 789-791; Angew. Chem. Int. Ed. 2020, 59, 2-9).

In the paper, our purpose is to detect cell samples, and we used DNA origami substrates to demonstrate our method. Still, all these supplementary experiments well

confirm the performance of our proposed method and help improve our paper. Now a related explanation about these results was added in the revised manuscript as below:

“ We also designed different two-dimensional DNA origami structures to verify TALKING. It is well known that nanoscale distances between different sites on DNA origami are easily controlled and designed. Firstly, we designed three types of barcoding probes surrounding the walking probe with about 7 nm radius distance as an example. These DNA probes were attached on the origami by DNA hybridization, and the origami was fixed on a coverslip (Figure 1C). Atomic force microscopy was used to confirm our DNA origami structure (Figure 1C, bottom left). The fluorescence signals of three channels overlapped (Figure 1C and S2A), which was consistent with our results in Figure 1B. In contrast, negative controls without probes or with mismatched probes presented very low fluorescence response (Figure S2A), indicating negligible false positive signals on the origami substrates. Notably, substantial heterogeneity in spot fluorescence overlap was observed. Thus, we quantified the number of spots with fluorescence overlap by five experimental replicates (Figure S2A). A portion of spots missed one or two of all three fluorescence signals, indicating the false negative level of our method on this origami substrate. This false negative result may be ascribed to both the limited efficiencies of reactions on coverglass and the failure of the formation of DNA origami substrates. It is well known that molecular reactions such as DNA hybridization and enzymatic reactions are of low efficiency at water/glass interface compared to in solutions. DNA hybridization on DNA origami is also restricted by strong electrostatic repulsion. The yield of DNA origami structures rarely reaches 100%, mainly depending on the complexity of design and experimental conditions. The failure of fluorescence overlap of spots in Figure 1A and 1B can be explained similarly. Notably, some spots presented relatively large sizes, which may

result from non-uniform RCA efficiency at this interface or the aggregation of multiple copies of DNA substrate molecules.

We further investigated the distance effect on proximity recognition of our method. DNA origami structures were designed to attach one barcoding probe with different separation distances (about 7 nm, 14 nm, or 22 nm) to the walking probe or target site (Figure 1D). The number of spots were reduced as the increase of separation distances. The normalized spot counts of the samples of other distances to that of 7 nm were shown in the bottom right panel of Figure 1D. In principle, the maximum recognition distance could reach the sum of the length of barcoding probe and that of walking probe. However, the stretching of flexible DNA may be restricted at water/glass interface or by strong electrostatic repulsion from DNA origami. Increasing the length of probes may extend the recognition distance. Here we used longer walking probes containing an additional spacer (15 nt or 30 nt) to investigate this issue. As shown in Figure S2B, longer walking probes increased the signal response at long-distance detection sites. This result confirmed the probe length-determined recognition range of our method. Therefore, our method achieved one-to-many proximity recognition and indexing performance. These results may enable the exploration of combinatorial patterns of diverse chromatin modifications in the same nanoenvironments.”

Figure S1. Other images of the experiments of *in vitro* Cell-TALKING performed on coverglass. (A) and (B) are corresponding to Figure 1A and 1B, respectively. Five repeated experiments were performed for statistical analysis of spot count of randomly selected images. The “spot overlap ratio” means the proportion of the spot count of one kind (fluorescence overlaps or not) in total number of all kinds. The letters B, G, and R indicate spots of blue, green, and red, respectively. There are some spots that fail to overlap two or three fluorescence signals, indicating the false negative response of our method on this DNA substrate on glass. It is discussed in the main text.

Figure S2. Performance of *in vitro* Cell-TALKING on DNA origami substrates. (A) Upper, the actual walking mechanism. Two copies of three barcoding probes (BPs) surround the blocked walking probe (WP) with about 7 nm radius distance on one origami molecule. During the nicking/walking process, the nicking enzyme firstly cuts the blocker sequence in the blocked WP duplex. Then the walking probe molecule in this duplex can be isothermally released to hybridize one BP molecule, forming a DNA duplex as the substrate for nicking enzyme. After cutting this BP molecule in the duplex, the walking probe molecule will walk to and hybridize another BP molecule. In this way, all these BP molecules can be cleaved and the newly generated 3'-OH ends can induce RCA reaction. Middle, analysis of false negative response of our method. The images

are corresponding to Figure 1C. Bottom, investigating the false positive response by several negative controls. There are some spots that fail to overlap two or three fluorescence signals, indicating the false negative response of our method on this DNA origami substrate. It is discussed in the main text. (B) Investigating the probe length-determined recognition range of our method using different walking probes and DNA origami substrates. Five repeated experiments were performed for statistical analysis of spot count of randomly selected images. The “normalized spot count” means the spot count of one sample relative to that of the given one (e.g., the positive control (IV) or the sample using walking probe containing 0 nt spacer).

Fig. 1D is unclear. Is it the case that the maximum distance that should be spannable is 7nm? In that case, how do we interpret the “leaky” positive signals detected at distances of 14 and 22 nm?

If, on the other hand, we are misinterpreting the claim, and it is indeed possible by design for the walking probe to reach 22 nm, then please rewrite for clarity.

In either case, then we recommend the authors should use experimental replicates of this DNA origami model to quantitatively characterize the tradeoff between distance and signal. Otherwise, in light of the admitted limitations of the microscopy resolutions used in the paper, it is hard to interpret data from downstream experiments.

Response: Thanks for the valuable comments. In fact, nanoscale distances between different sites on DNA origami are easily controlled and designed, and the minimum distance is roughly about 5-10 nm depending on different origami designs or locations. In our work, we first designed multiple barcoding sequences surrounding one walking probe with about 7 nm radius distance as an example to simulate a nanoenvironment containing diverse modification sites (Figure 1C and S2A). Then we designed barcoding probes with different separation distances (about 7 nm, 14 nm, or 22 nm) to the walking probe to investigate the recognition range of our method.

In principle, the maximum recognition distance of our method could reach the

sum of the length of barcoding probe and that of walking probe. Based on 10 bp dsDNA spacing of ~ 3.4 nm, the lengths of these two probes can be roughly determined as about 20 nm and 25 nm, respectively. However, the stretching of flexible DNA can be restricted at water/glass interface, by DNA secondary structure and/or by strong electrostatic repulsion from DNA origami. As shown in Figure 1D, the spot counts of long-distance detection sites (14 nm and 22 nm) are much lower than that of 7 nm detection site. The numbers of spot were summarized by five experimental replicates to characterize the tradeoff between distance and signal. This result accords with above-mentioned explanation.

To further investigate this issue, we supplemented related experiments. The results were shown in S2B. We used other walking probes containing a longer spacer (additional 15 nt or 30 nt) to probe the long-distance detection site (22 nm). We found that long walking probes increased the signals at 22 nm detection site. It demonstrated the probe length-determined recognition/detection range of our method. A related description has been added in main text of revised manuscript as below:

“We further investigated the distance effect on proximity recognition of our method. Other DNA origami structures were designed to attach one barcoding probe with different separation distances (about 7 nm, 14 nm, or 22 nm) to the walking probe or target site (Figure 1D). The number of spots were reduced as the increase of separation distances, and the normalized spot counts of other distances to that of 7 nm were shown in the right bottom panel of Figure 1D. In principle, the maximum recognition distance could reach the sum of the length of barcoding probe and that of walking probe. However, the flexible DNA stretching may be restricted both at water/glass interface and by strong electrostatic repulsion from DNA origami. Increasing the length of probes can extend the recognition distance. Here we used longer walking probes containing an additional spacer (15 nt or 30 nt) to investigate this issue. As shown in Figure S2B, longer walking probes increased the signal response at long-distance detection sites. This result confirmed the probe length-determined recognition range of our method. Therefore, our method overcame the limitation of one-to-one DNA proximity recognition and achieved one-to-many indexing performance. These results may enable the exploration of combinatorial patterns of diverse chromatin modifications in the same nanoenvironments.”

Figure 1. Proof-of-principle experiments of in vitro Cell-TALKING performed on DNA duplex-modified coverglass (A and B) and on DNA origami (C and D). The scale bars of images are 25 μm and 20 μm in Fig. 1A, 10 μm in Fig. 1B, 20 nm and 5 μm in Fig. 1C, 15 μm in Fig. 1D, respectively. The BP and WP indicate the barcoding probe and walking probe, respectively. Five repeated experiments were performed for statistical analysis of spot count of randomly selected images.

Negative controls for the fixed cell experiments in Fig. 3.

What we mean here is to repeat the experiments, except with the blocked labels missing and/or with scrambled primer sites. This will confirm that the signals we observe are not spurious, and require all components to be present.

Another useful negative control we recommend is to first flush in labels missing the proper blocked DNA oligos and allow them to “label” or “occupy” all sites of interest. Then flow in proper labels for that site. None (or very few) should attach,

and hence yield almost no signal. This would be a control against spurious labeling.

Response: Thanks for the valuable comment. We have supplemented related experiments, and the results were shown in Figure S4B and S4C. All these revisions help improve our paper.

For the final negative control recommended by the reviewer, we explain below. In our design, the barcoding probes for different DNA modifications are modified at 3' ends with phosphorothioated bases and locked nucleic acid for the inhibition of both 3' to 5' digestion and 5' to 3' extension by phi 29 DNA polymerase. It means that these barcoding probes are locked at 3' ends and disabled to induce RCA without nicking reaction in Cell-TALKING. In addition, the walking probe for histone PTM is blocked at its 3' end by hybridizing a complementary ssDNA blocker. The 3' end sequence of the walking probe is also complementary to that of the barcoding probes. This ssDNA blocker/walking probe duplex (named blocked walking probe) prevents the hybridization between walking probe and barcoding probes before nicking reaction in Cell-TALKING, which avoids non-specific signal response from the modification sites outside the nanoenvironment. If we use the control of barcoding probes without 3' end lock, all the modification sites (within or outside the nanoenvironment) can induce RCA catalyzed by phi 29 DNA polymerase without the requirement of nicking reaction. If we use the control of walking probe without 3' end ssDNA blocker, all the modification sites (within or outside the nanoenvironment) can capture this ssDNA walking probe via their 3' locked barcoding probes, and these barcoding probes will be nicked during the nicking reaction and induce RCA. Thus both two controls can generate strong false positive signals.

We also supplemented related negative controls in Figure S4 to investigate the spurious labeling. Among them, the "Mismatched probes" negative control used a circularized padlock that can not hybridize to all three barcoding probes, which fails to induce RCA. And the "Pre-block modification sites" control firstly labeled all modification sites with blocked duplex barcoding probes, and then with single-stranded barcoding probes. The blocked duplex barcoding probes were prepared by the hybridization of the complementary oligo to the single-stranded

barcoding probes. They can not hybridize to their circularized padlocks and fail to induce RCA. This negative control can indicate that almost all sites were labeled and crosslinked with DNA probes by the first labeling processes, and no residual sites were detected by second labeling processes.

Figure S4. Specific cell imaging using Cell-TALKING. (A) Scheme of sequential labeling of 5hmU, 5hmC and 5fU followed by the histone PTM in cells. The blocked walking probe is a DNA hybrid duplex that can be captured by the DNA-crosslinked secondary antibody. The detail is described in the protocol of Cell-TALKING as below. (B) Selected cell images for several negative controls and one positive sample. The nanoenvironment around histone H3K27ac in MCF-10A cells was detected here. Scale bar, 10 μ m. (C) The spot counts of three combination

sites of single cells (cell numbers are 50). The “Non-walking probe” negative control used the Non-walking probe instead of the walking probe. The Non-walking probe can not hybridize to all three barcoding probes, which fails to induce the DNA nicking reaction and walking process. The “Mismatched probes” negative control used a circularized padlock that can not hybridize to all three barcoding probes, which fails to induce RCA. And the “Pre-block modification sites” control firstly labeled all modification sites with blocked duplex barcoding probes, and then with single-stranded barcoding probes. The blocked duplex barcoding probes were prepared by the hybridization of the complementary oligo to the single-stranded barcoding probes. They can not hybridize to their circularized padlocks and fail to induce RCA. This negative control can indicate that almost all sites were labeled and crosslinked with DNA probes by the first labeling processes, and no residual sites were detected by second labeling processes. Similar results are shown in Figure S3.

In multiple figures & analyses, such as in Fig. 3C, authors apply cross-correlations to one-dimensional cross-sections of puncta to quantify correlations. While is this valid in principle, it is less powerful and informative than using two-dimensional metrics for puncta overlap.

Puncta colocalization analysis tools are regularly used in cell biology to look at e.g. colocalization of RFP- & GFP-labeled proteins. There are multiple metrics available. Tools such as ImageJ should have modules available.

Response: Thanks for the valuable suggestion. In our paper, we used confocal microscopy for proof-of-principle study, despite its limited resolution. As described in the section of “Imaging and data analysis”, we imaged the cells in 3D with the z-stack at intervals of 0.5 μm , and the resulting z-stack images were combined to a single one by maximum intensity projection using LAS X Version of the microscope. These maximally projected images of cells can be used for the extraction of single-cell fluorescence intensity and spot count by Matlab, ImageJ plug-in or any other

image-processing algorithms. We do this processing following the instructions of the softwares, and it is consistent with those of previous studies (Nature Protocols, 2012, 7, 408-419; Nature Protocols, 2013, 8, 1743-1758; Chem, 2018, 4, 1-14.).

We also roughly depicted the fluorescence overlap of spots according to a large distance of hundreds of nanometers in Figure 3C. As suggested by the reviewer, here we supplemented the Z-stack 3D image analysis of the image used in Figure 3C as an example to present possible signal overlap in Z-stack 3D. The results were shown in the revised Figure 3C and Figure S7. We can see that the spots highlighted in circle are overlapped in two or three fluorescence channels with the resolution of hundreds of nanometers. Notably, some overlapped spots in maximally projected images are not overlapped in 3D. The data in Figure 3C are conceptually and roughly presented based on above-mentioned conditions, which has also been described in the main text of the manuscript. Thus we didn't summarize numbers of the spots that were overlapped in 3D or 2D again. Especially, strict and systematic experiments are required to verify above results.

Super-resolution microscopy is a better choice than confocal microscopy to detect the colocalization of two or three targets. But it often suffers from the registration errors or positioning errors from sample drift in multichannel imaging and low imaging throughput, and its performance can vary considerably in optically complex samples. In principle, other DNA proximity recognition techniques similar to that used in our method can be developed to directly detect the nanoscale colocalization of more than three given targets. For example, we can design a complex DNA structure or substrate that will only be formed in the presence of all the DNA recognition probes within the same nanoenvironment. And this DNA formation product can only trigger one signal that indicates the colocalization of those targets. Such a design and study may be another interesting work. In summary, all these revisions help improve our paper, and we hope them meet the requirements.

Figure 3. Exploring different nanoenvironments of chromatin modifications in single cells. (A) Selected merged cell images for five histone PTMs. The red, green and blue fluorescence spots each indicated the combination sites of 5hmU/PTM, 5fU/PTM and 5hmC/PTM. The scale bar is 5 μm. (B) The spot percentages of three combination sites of single cells in five samples (cell numbers = 52, 34, 32, 43 and 38, respectively.). (C) Analysis of the overlap or cluster of different spots.

Figure S7. 3D overlap analysis of the spots from different fluorescence channels in the image used in Figure 3C. The images were acquired in 3D with the z-stack at intervals of 0.5 μm . The spots highlighted in circle are overlapped in three or two fluorescence channels with the spatial resolution of hundreds of nanometers.

Minor revisions:

Overall language and explanations need to be rewritten more clearly before publication.

Response: Thanks for the valuable comment! We have clearly revised the main text, and requested the insights of two experts to improve the manuscript.

Need a clearer diagram and explanation of the actual walking mechanism in Fig 1. This is a key concept & figure.

Response: Thanks for the valuable comment! As suggested, we have drawn an actual walking mechanism in Figure S2A.

Fig. 1D: Please clarify what “Relative spot (%)” means, and how it’s calculated. We have an idea of what the authors mean, however clarity here would help.

Make clear in Fig. 1 or main text description what the locked base actually is. Per supplement it is iXNA_G.

Response: Thanks for the valuable comments! We revised the ‘Relative spot (%)’ to ‘normalized spot count’ in related figures by five repeated experiments. It means the spot count of one sample relative to that of the given one. And the explanation can be seen in the legends of Figure 1D, S1 and S2.

For the locked base, it is a base of locked nucleic acid (LNA) and its chemical structure is shown below according to Integrated DNA Technologies (<https://sg.idtdna.com/pages/technology/custom-dna-rna/locked-nucleic-acids>):

And the word “3’ locked” in Scheme 1 and Figure 1 is used to indicate the 3’ terminal sequence containing several phosphorothioated bases and a base of locked nucleic acid (LNA), which can inhibit both 3’ to 5’ digestion and 5’ to 3’ extension. We have made relevant revisions in the Table S1 and the main text.

“This 3’ terminal sequence contains several phosphorothioated bases and a base of locked nucleic acid (LNA) for the inhibition of both 3’ to 5’ digestion and 5’ to 3’ extension, which is depicted as 3’ locked in figures.”

Explicitly point out if this walker design has been published before by authors themselves or others, or is it completely novel? If it is not completely novel, please provide a summary of differences and/or citations.

Response: Thanks for the valuable comment! In fact, the design mechanism of the walker involved in our method is not completely novel. It is similar to those reported by us or others that are powered by enzymes or DNA strand displacement. Here this one using nicking enzymes is simpler and more robust than those using DNA repair enzymes, exonucleases or RNase H. And all DNA walkers have not been utilized to probe cellular nanoenvironments before our paper. We have added a relevant explanation and citations in the main text of the revised manuscript as below:

“Notably, the mechanism of the DNA walker used in our method is similar to others³¹⁻³⁶, especially the enzyme-powered ones^{31, 33-35}. Among them, the design using nicking enzymes³⁴ is simpler and more robust than those using DNA repair enzymes, exonucleases or RNase H. In principle, other walkers may also be utilized to perform similar function in our method.”

Fig. 3B and similar figures need to clarify what “Spot Percentage (%)” means and how it’s quantified. The description as-is is not completely transparent.

Response: Thanks for the valuable comment! We have added a clear description in the main text of the revised manuscript as below:

“The averaged spot percentages of three combination sites were shown in Figure 3B. This percentage means the proportion of the spot count of one combination modification in that of all three in the same single cells.”

REVIEWER COMMENTS

Reviewer #1 (Remarks to the Author):

While the authors improved the manuscript in this revision, I still have major concerns regarding the novelty and utility of the method and the extent to which the authors applied it to reveal new biology. Specifically:

1. In this manuscript, the reviewers did not see significantly increased novelty (either scientifically or technically) in comparison with the other similar papers published by the same group (e.g., *Angew Chem. Int. Ed. Engl.*, doi: 10.1002/anie.202011172; *J. Am. Chem. Soc.*, 142, 2889-2896). Instead of just visualizing the nanoenvironments, more pathophysiological significance should be explored. If not, this manuscript should be transferred to a chemistry or methodology journal.
2. In the point-to-point response, the authors claimed that "Notably, we can't validate whether the detected spatial proximity PTMs (or sites) locate on the same tail (or the same genomic region for DNA marks) or not. And as far as we know, there is no existing methods to determine this issue." In fact, there are papers published regarding the development of engineered multivalent sensors to detect coexisting histone modifications in cells (e.g., *Cell Chem. Biol.*, 2018, 25, 51-56).
3. Although the authors have added more details regarding the probe synthesis, the synthetic methodology of ATP- γ -alkyne was not initially performed by the authors. The original paper (*Chem. Commun.*, 2014, 50, 1861-1863) should be cited. The probe, ATP- γ -alkyne (one carbon shorter than the one reported in the *Chem. Commun.* paper), is a new compound, thus comprehensive structural data should be provided, including NMR.

Reviewer #2 (Remarks to the Author):

The authors have adequately addressed my concerns in the revised manuscript. Most notably, authors have addressed the specificity of their methods by a number of control experiments and additional discussion. I fully support publication of this manuscript in *Nature Communications*.

Reviewer #3 (Remarks to the Author):

The necessary and requested controls we requested have been added.

We have no further concerns and appreciate the clarifications and changes in response to our comments.

Point-by-point response to the reviewers' comments for NCOMMS-20-27973A (Title: Cellular Macromolecules-Tethered DNA Walking Indexing to Explore Nanoenvironments of Chromatin Modifications)

Reviewer #1

While the authors improved the manuscript in this revision, I still have major concerns regarding the novelty and utility of the method and the extent to which the authors applied it to reveal new biology. Specifically:

*1. In this manuscript, the reviewers did not see significantly increased novelty (either scientifically or technically) in comparison with the other similar papers published by the same group (e.g., *Angew Chem. Int. Ed. Engl.*, doi: 10.1002/anie.202011172; *J. Am. Chem. Soc.*, 142, 2889-2896). Instead of just visualizing the nanoenvironments, more pathophysiological significance should be explored. If not, this manuscript should be transferred to a chemistry or methodology journal.*

Response: Thanks for the comment. The reviewer mainly raised two concerns:

1) Novelty of current work compared our previous works;

We are very sorry for this confusion. To highlight the novelty of this work, we made a detailed comparison to our previous two papers published in *JACS* and *Angew. Chem.* In our *JACS* paper (Scheme A), we proposed a chemoenzymatic labeling method to distinguish 5hmU and 5hmC, two structurally similar DNA base modifications. In our *Angew. Chem.* paper, we proposed a pairwise proximity-differentiated DNA recognition mechanism to investigate the spatial proximity of two base modifications (5fC and 5hmC as an example, one-to-one proximity) in cells. The proposed method is partly similar to existing proximity ligation assays (PLAs). But it overcame the limitation of simultaneous detection of both proximal sites and non-proximal sites compared to PLAs.

Besides the simple proximity between two biomolecules (one-to-one proximity), there are complicated biomolecule proximity in molecularly crowded cellular environments, such as many proximal biomolecules surrounding a single target biomolecule (one-to-many nanoenvironments). **However, similar to existing PLAs, our pairwise proximity-differentiated method cannot detect one-to-many nanoenvironments (Scheme B). It is because only a single proximity record (PR) is yielded from each probe. Even its multiplex version can only detect different**

one-to-one proximities. But these proximity information are isolated in unconnected (spatially independent) pairs which are not the one-to-many nanoenvironments. The major value of our current manuscript (NCOMMS-20-27973) lies in developing a conceptually novel DNA walking indexing mechanism to probe one-to-many nanoenvironments. This mechanism, called Cell-TALKING in this work, is shown in III section of Scheme B. One walking probe (WP) molecule (for individual PTM site or T site) was reused to continuously activate any nearby DNA barcoding probes (BPs, for the sites of multiple DNA modifications) via the reaction circles of DNA proximity walking indexing. In this way, a single WP yields many PRs. Multiplex RCA reactions were used to visualize these activated BPs within the nanoenvironments rather than 3' locked BPs outside the nanoenvironments. Especially, RCA are not the major value in our papers. The following schematic illustration is presented to clarify the major difference about novelty of our three papers.

Schematic illustration of the novelty of our three papers.

(A) The novelty of our JACS paper (J. Am. Chem. Soc., 2020, 142, 2889-2896).

(B) Comparison between existing PLAs (including our PLA in Angew. Chem. Int. Ed., doi: 10.1002/anie.202011172 and multiplex PLA) and our Cell-TALKING (this manuscript NCOMMS-20-27973). (I) Different mechanism of these methods and their different detection performance for one-to-many nanoenvironments. In PLA methods, the DNA probes are destructive and not reused, thus each probe can only yield a single proximity record (PR). In our

Cell-TAKING, a single WP molecule (for one T site) is recycled and reused to continuously activate many BP molecules (for multiple proximal sites), generating one-to-many proximity records.

(II) The detailed mechanism of typical PLA. Two different analytes are first labeled with respective ssDNA primers, and these primers each can hybridize one set of two split probes. During the reaction of DNA proximity ligation, the two nicks at pairwise proximity sites are both ligated, generating the circularized DNA barcode. The ssDNA primer probes are destructive in the proximity ligation reaction, and can not be reused to record another one-to-one proximity.

(III) The detailed mechanism of our Cell-TAKING (this manuscript NCOMMS-20-27973). Firstly, the sites of one histone PTM are labeled with a duplexed walking probe (WP), and the sites of different DNA modifications are labeled with respective ssDNA barcoding probes (BPs). These BPs are treated as the molecular tracks for the WP in our DNA walking system. Especially, different BPs have unique barcode sequences in the middle and have identical adapter (Ad) sequences at 3' end. Their 3' terminal sequence contains several phosphorothioated bases and a base of locked nucleic acid (LNA) to inactivate the BPs via the inhibition of both 3' to 5' digestion and 5' to 3' extension. On the other hand, the duplexed WP is sealed at 3' end with a complementary ssDNA blocker, which inhibits the unwanted hybridization of the BPs before nicking reaction. The 3' terminal sequence of the WP is designed to be complementary to the adapter sequences of the BPs. After the labeling reaction, a nicking enzyme is added to power the DNA walking indexing process. The ssDNA blocker in the duplexed WP is cleaved into two short DNA oligos by the nicking enzyme. It results in the unstable DNA hybridization between cleaved ssDNA blocker and ssDNA WP, which releases the ssDNA WP. Then this ssDNA WP can recognize and cleave any nearby BPs by multiple reaction circles of stable DNA hybridization, BP nicking, unstable (or dynamic) DNA hybridization and isothermal probes releasing. The cleaved BPs contain new 3'-OH ends instead of 3' locked ends of uncleaved BPs. These 3'-OH ends can activate the BPs via Phi 29 polymerase-catalyzed RCA reaction for cell imaging.

To clearly clarify the novelty of our current paper, we also revised the manuscript (Introduction and Discussion) and improved the Scheme 1 as below:

Revised part of Introduction:

“For the first time, we reported specific thiophosphorylation of 5hmU against its structurally similar analogues such as 5hmC and 5fU, and achieved differentiated imaging of 5hmU and 5hmC in the same cells.²⁰ Aforementioned imaging methods could be routinely expanded for parallel detection of three or more spatially independent modifications by designing orthogonal probe sets. Yet they are unable to probe spatial proximity relationships between two or more different modifications within nanoenvironments.”

Revised part of Discussion:

“As described in the introduction, the analysis of cellular nanoenvironments is quite different to both the detection of two proximity targets by PLA and even the independent, parallel PLA analysis. Very recently, we proposed a one-to-one pairwise proximity-differentiated DNA recognition mechanism based on the skillfully integrated design of an improved DNA proximity ligation and toehold-mediated strand displacement.⁴⁶ We used this mechanism to investigate the spatial proximity of two DNA modifications (5fC and 5hmC). It is able to successively encode 5fC/5hmC pairwise proximity sites, residual 5fC sites and residual 5hmC sites with respective circularized DNA barcodes as RCA templates. It overcame the limitation of current PLA methods that they can't detect the residual sites of two analytes. However, our pairwise proximity-differentiated mechanism can't probe one-to-many proximity within nanoenvironments because only a single proximity record is generated from each DNA probe. Even its multiplex version can only detect multiple one-to-one proximities, but these proximities are independent pairs rather than the one-to-many nanoenvironments. In order to overcome this challenge, we here designed our Cell-TALKING method based on DNA proximity walking indexing. But the Cell-TALKING method is unable to detect the analyte sites outside the nanoenvironments. A modified Cell-TALKING version combined with our aforementioned proximity-differentiated concept may be feasible to differentially detect the sites within and outside the nanoenvironments in the same cells. Furthermore, PTMs recognition via antibodies used in our method may be affected by epitope occlusion.⁴⁷”

1-2) A related work published during the second revision of our manuscript;

Especially, during the revision of our manuscript, Nature Nanotechnology online published an article about detecting one-to-many nanoenvironments of membrane proteins (Title: A DNA-nanoassembly-based approach to map membrane protein nanoenvironments; link: <https://www.nature.com/articles/s41565-020-00785-0>). In the Nature Nanotechnology article, the authors assembled a DNA NanoComb to probe spatial proximity of one-to-many membrane proteins (three or four other proteins surrounding one Her2 protein), as shown in the following scheme.

Schematic of the NanoDeep method to map membrane protein nanoenvironments by DNA nanoassembly and sequencing (Nature Nanotechnology, 2020,

<https://doi.org/10.1038/s41565-020-00785-0>

The goal of detecting one-to-many nanoenvironments by their method is similar to that of our NCOMMS-20-27973A paper, but these two methods are significantly different in the mechanism of DNA proximity recognition and barcoding. Their method used pre-assembled DNA nanoassemblies with multiple DNA barcodes to translate organization information of membrane proteins into DNA sequencing readouts. They used DNA next generation sequencing rather than DNA hybridization fluorescence imaging to detect the DNA products of proximity recognition reaction. They required 400,000 cells per sample for DNA sequencing analysis and lacked subcellular distribution information in comparison with single-cell imaging analysis. Though DNA sequencing can achieve very high detection throughput than cell fluorescence imaging, single-cell sequencing analysis of cellular nanoenvironments may be still an open challenge. In addition, their pre-assembled DNA nanoassemblies own a relatively large size of about 34 nm in comparison with common DNA probes. The large size may hinder its diffusion in molecularly crowded intracellular environments especially cell nucleus. Thus their method is perhaps not appropriate for the detection of analytes in intranuclear settings. We have made a comparison between their method and our Cell-TALKING method in the revised manuscript, as shown below:

Revised part of Discussion:

“Recently, microscopy-by-sequencing as an emerging technique, has been established to probe the spatial organization of biomolecules inside cells.^{55,56} It can avoid the optical resolution limitation and provide a global spatial map for a complex molecular population by using DNA proximity recognition and high-throughput sequencing. Inspired by these advancements, our Cell-TALKING method may be integrated with sequencing replacing fluorescence imaging for the profiling of accurate nanoscale organization of even tens of or hundreds of components. Especially, during the revision of our manuscript, one method directed at the same goal, NanoDeep, was published.⁵⁷ NanoDeep used pre-assembled DNA nanoassemblies with multiple position barcodes to translate spatial organization information of membrane proteins into DNA sequencing readouts. This method can probe the nanoenvironments of one-to-many membrane proteins (three or four other proteins surrounding Her2 protein), but it required hundreds of thousands of cells as start materials.

Its mechanism of DNA proximity recognition and barcoding is quite different to that of our Cell-TALKING method. Notably, the pre-assembled DNA nanoassemblies own a size of about 34 nm. This relatively large size may hinder its application in molecularly crowded intracellular environments especially cell nucleus. And single-cell sequencing analysis of cellular nanoenvironments may be still a common challenge. Another issue is that Cell-TALKING can't record the 'many-to-many' molecular combinations or relationships. It is still challenging to probe any or a complete set of combinatorial patterns between groups of molecules in the same nanoenvironments. Conceptually, it requires to record nondestructive and continuous interaction of any two adjacent DNA probes."

2) Significance of current work in a broader field of biology;

Actually, the significance is supported by many previous works including above Nature Nanotechnology ref and the suggested one (Cell Chem. Biol., 2018, 25, 51-56) in the second comment. These works provide versatile methods to detect proximity of proteins or epigenetic modifications, and they have the potential to highlight the importance of spatial organization of different biomolecules in biology. However, they cannot achieve what have been done here as discussed above. Besides chromatin modifications, the method proposed by our current work can be easily modified to detect the nanoenvironments of other biomolecules such as transcription factors or membrane proteins. Similarly, our method also provides a tool for understanding the importance of cellular nanoenvironments. Nevertheless, it is still an open question to reveal the critical roles of nanoenvironments of chromatin modifications in the regulation of gene function and even in pathophysiological processes.

2. In the point-to-point response, the authors claimed that “Notably, we can’t validate whether the detected spatial proximity PTMs (or sites) locate on the same tail (or the same genomic region for DNA marks) or not. And as far as we know, there is no existing methods to determine this issue.” In fact, there are papers published regarding the development of engineered multivalent sensors to detect coexisting histone modifications in cells (e.g., *Cell Chem. Biol.*, 2018, 25, 51-56).

Response: Thanks very much for pointing out the improper description about the detection of coexisting histone modifications in our point-to-point response. The paper (*Cell Chem. Biol.*, 2018, 25, 51-56) estimated the average separation distances of two coexisting histone modifications from chain statistics. The value is about 4 nm when they are on the same histone tail, and it is about 10 nm when they are on two different histone tails in a nucleosome. The minimum distance between two nucleosomes is about 15 nm. Considering the large sizes of antibodies (about 150 kDa, approximately 15 nm), antibodies-based methods may not accurately determine whether two coexisting histone modifications locate on the same tail or not. And they require cell lysis or fixation and are thus not applicable for dynamic experiments in living cells. To circumvent these problems, that paper report genetically encoded chromatin-sensing multivalent probes for selective detection of bivalent chromatin in living cells. The authors engineered a series of fusion proteins containing a fluorescent reporter and two reader domains, each specific for a histone PTM, connected by linker sequences. They optimized the length of linker sequences to change the size of fusion proteins to meet the distance between two coexisting histone modifications on the same tail. They selectively detect two combinatorial PTMs on the same histone tail by those fusion protein sensors with small sizes.

Inspired by this paper, we investigated the publications about antibodies with smaller sizes. We found that the emerging nanobodies (*Angew. Chem. Int. Ed.* 2018, 57, 2314-2333; *Front. Cell. Neurosci.*, 2020, doi.org/10.3389/fncel.2020.573278; *FEBS Open Bio*, 2015, 5, 779-788) with the size of about 4 nm (12-30 kDa) may be also applicable to accurate detection of the coexisting or proximity of histone PTMs in fixed cells. In our manuscript, we probed the nanoenvironments of one-to-many nanoenvironments (three DNA modifications around one histone PTM) by our proposed method, and we didn’t detect the coexisting of two or more histone PTMs. **In order to better explain aforementioned questions and improve our manuscript, we have added a related description in the Discussion section of the revised manuscript as below:**

Revised part of Discussion:

“Furthermore, PTMs recognition via antibodies used in our method may be affected by epitope occlusion.⁴⁷ The specific binding of antibodies to a single histone PTM could be obfuscated by any neighboring PTMs at the same tail. This steric occlusion effect is a common issue for PTMs recognition, especially on highly modified histones. It requires to generate more specific antibodies against histones containing various modification valences.⁴⁸ On the other hand, the detected combinatorial chromatin modifications by our method may locate on different genomic regions and even on different chromatins ascribed to complicated chromatin interaction in cell nucleus. Due to the large size of antibodies (about 150 kDa, approximately 15 nm), antibodies-used methods may not accurately determine whether two coexisting PTMs locate on the same histone tail or not. The separation distance of two PTMs on the same histone tail is about 4 nm.⁴⁹ Genetically encoded multivalent protein sensors with small sizes have been reported to selectively detect combinatorial PTMs on the same histone tail.⁴⁹ In addition, the emerging nanobodies with the size of about 4 nm (~ 12-30 kDa) may be also promising tools.⁵⁰⁻⁵²”

3. *Although the authors have added more details regarding the probe synthesis, the synthetic methodology of ATP- γ -alkyne was not initially performed by the authors. The original paper (Chem. Commun., 2014, 50, 1861-1863) should be cited. The probe, ATP- γ -alkyne (one carbon shorter than the one reported in the Chem. Commun. paper), is a new compound, thus comprehensive structural data should be provided, including NMR.*

Response: Thanks for the valuable comment! As suggested, we have cited two original papers (J. Org. Chem. 2012, 77, 10450-10454; Chem. Commun., 2014, 50, 1861-1863) about the synthetic methodology of ATP- γ -alkyne in the revised manuscript. We used a synthetic methodology similar to those of these two papers. The structure of our ATP- γ -alkyne is a little different. We have provided its structural characterization including NMR and IR shown in the Figure S3 and Supplementary Materials of the revised Supplementary Information as below:

Figure S3. The structural characterization of ATP- γ -alkyne. See more details in the following Supplementary Materials.

“Chemical synthesis and characterization of ATP- γ -alkyne. Adenosine 5'-triphosphate (ATP) disodium salt (Sigma Aldrich, cat. no. 10519979001) and propargylamine (Sigma Aldrich, cat. no. P50900) were used to synthesize ATP- γ -alkyne. ATP disodium salt (4.9 mM) and propargylamine (245 mM) were dissolved in water. And then the pH was adjusted to 6.0. 490 mM EDC-HCl (Sigma Aldrich, cat. no. E6383) was added to the above solution, and the mixture was stirred for 24 h at room temperature. Then the pH was adjusted to 8.5. After evaporating, the light yellow viscous products was purified with silica gel column chromatography to get the white solid ATP- γ -alkyne. ATP: HRMS (ESI, negative mode) for $C_{10}H_{15}N_5O_{13}P_3$, $[M+H]^-$: 505.98847 (calculated), 505.98488 (found). ATP- γ -alkyne: HRMS (ESI, negative mode) for $C_{13}H_{18}N_6O_{12}P_3$, $[M+H]^-$: 543.02010 (calculated), 543.01591 (found). All the mass spectrometry analyses in this work were performed on a Waters I-Class Vion IMS Qtof with electrospray ionization.

NMR Spectra were recorded on a Bruker AVANCE III HD 600 MHz machine. The deuterated solvent in this work was D_2O . ^{31}P NMR spectra are referenced to H_3PO_4 as an external standard. ^{13}C NMR spectra are referenced to tetramethylsilane (TMS) as an external standard. Chemical shifts (δ) are quoted in parts per million (ppm) and coupling constants (J) are measured in hertz (Hz). The following abbreviations are used to describe multiplicities: s=singlet, d=doublet, t=triplet, m = multiplet. The spectra are shown in Figure S3.

1H NMR (600 MHz, D_2O): δ (ppm) = 8.50 (s, 1H, 2-H), 8.18 (s, 1H, H8), 6.10 (d, $J=6.09$ Hz, 1H, H1'), 4.80 (m, 1H, overlapping with HOD, H2'), 4.62-4.55 (m, 1H, H3'), 4.38 (m, 1H, H4'), 4.24 (m, 2H, H5'), 2.79-2.65 (m, 3H, H12, H14).

^{13}C NMR (150 MHz, D_2O): δ (ppm) = 155.4 (C6), 152.7 (C2), 149.0 (C4), 139.8 (C8), 118.4 (C5), 86.7 (C1'), 83.9 ((d, $J_{PC}=9.0$ Hz, C4'), 82.3 (C13), 74.2 (C2'), 70.3 (C3'), 70.2 (C14), 65.2 (d, $J_{PC}=5.6$ Hz, C5'), 23.5 (C12).

^{31}P NMR (243 MHz, D_2O): δ (ppm) = -2.75 (d, $J=20.5$ Hz, γ), -11.37 (d, $J=19.1$ Hz, α), -22.63 (t, $J=19.6$ Hz, β).

IR measurements were performed on Bruker VERTEX70 Micro-Infrared Spectroscopy equipped with a diamond-ATR setup. IR (ATR): $\tilde{\nu}$ (cm^{-1}) = 3249, 2976, 2680, 2472, 2119, 1620, 1562, 1481, 1384, 1338, 1261, 1164, 1082, 1047, 881, 599.

”

REVIEWERS' COMMENTS

Reviewer #1 (Remarks to the Author):

The authors have addressed all my comments.